# CPGAN: Towards a Better Global Landscape of GANs

## Abstract

GANs have been very popular in data generation and unsupervised learning, but our understanding of GAN training is still very limited. One major reason is that GANs are often formulated as non-convex-concave min-max optimization. As a result, most recent studies focused on the analysis in the local region around the equilibrium. In this work, we perform a global analysis of GANs from two perspectives: the global landscape of the outer-optimization problem and the global behavior of the gradient descent dynamics. We find that the original GAN has exponentially many bad strict local minima which are perceived as mode-collapse, and the training dynamics (with linear discriminators) cannot escape mode collapse. To address these issues, we propose a simple modification to the original GAN, by coupling the generated samples and the true samples. We prove that the new formulation has no bad basins, and its training dynamics (with linear discriminators) has a Lyapunov function that leads to global convergence. Our experiments on standard datasets show that this simple loss outperforms the original GAN and WGAN-GP.

## 1 Introduction

Generative Adversarial Networks (GANs) (Goodfellow et al., 2014) have been one of the most popular methods for generating data. The original GAN minimizes the loss $\phi(p_{\mathrm{g}}, p_{\mathrm{data}}) = \max_D h_D(p_{\mathrm{g}}, p_{\mathrm{data}})$, where $h_D$ is a binary classification loss that depends on the discriminator $D$. To justify the loss, Goodfellow et al. (2014) prove two theoretical results: first, for a given $p_{\mathrm{g}}$, the outer function $\phi(p_{\mathrm{g}}, p_{\mathrm{data}})$ is the Jenson-Shannon (JS) distance (minus a constant); second, the outer function is convex in $p_{\mathrm{g}}$, so a gradient descent method on $p_{\mathrm{g}}$ converges to the global minimum.

In order to further understand the training behavior of GANs, there has been a surge of interest in min-max optimization or games. One major challenge is the question about which problems to analyze. The first candidates are the two-person matrix game $\min_\psi \max_\theta \psi^T A \theta$ or general convex-concave problems. But the GAN formulation is not a convex-concave problem. As stated by Daskalakis & Panageas (2018), our knowledge of min-max optimization in non convex-concave settings is "very limited." Therefore, existing results often only analyze local stability or local convergence (Daskalakis et al., 2017; Daskalakis & Panageas, 2018; Azizian et al., 2019; Gidel et al., 2018; Mazumdar et al., 2019; Yazıcı et al., 2018; Jin et al., 2019; Sanjabi et al., 2018). Local analysis can tell us the behavior of the algorithms near the desired point, but how do we know that the algorithm will arrive there and not at some highly sub-optimal points? Answering this question requires a global landscape analysis or a study of global training dynamics.

There are some attempts on a global analysis of GANs. Mescheder et al. (2018) and Feizi et al. (2017) analyzed the simplest setting: the true data distribution is a single point and a single Gaussian distribution respectively. In these cases, the training dynamics of some GANs can converge to the globally optimal solution. Thus at least in the simplest single-mode setting, there is evidence that the global landscape of (some) GANs is compelling. However, it remains unclear whether those GANs have similarly nice properties in the multi-modal setting. Again, the difficulty here is that the practical GAN formulation is not convex-concave, even for simple discriminators/generators. These works and other related ones are reviewed in more detail in Appendix B.

**Our contributions.** In this work, we make a step towards understanding the global behavior of GAN training, under multi-modal settings. We focus on analyzing GANs for learning a multi-point

distribution. This is a finite-sample version of the problem analyzed by Goodfellow et al. (2014), and a multi-mode extension of the Dirac-GAN problem analyzed by Mescheder et al. (2018). We approach the problem from two perspectives. First, we consider the min-max optimization problem and analyze the landscape of the outer problem assuming powerful discriminators. This perspective was used by Goodfellow et al. (2014). In contrast, as optimization variables we use the samples $Y$ instead of the probability density $p_{\mathrm{g}}$. Second, we consider the training dynamics of the game formulation assuming linear discriminators. Our contributions are summarized as follows.

- For the original GAN, we prove that the outer-minimization problem has exponentially many sub-optimal strict local minima. Each strict local minima corresponds to a mode-collapse situation.

- For both the original GAN and WGAN-GP, we prove that with linear discriminators, the training dynamics cannot escape from mode-collapse.

- We propose a new GAN formulation called CP-GAN (CoupleGAN) that enjoys nice global properties. From the first perspective, we prove that the outer-minimization problem of CP-GAN has no bad strict local minima, improving upon the original GAN. From the second perspective, we prove that with linear discriminators, the training dynamics of CP-GAN can escape from mode-collapse; in addition, it has a global Lyapunov function that permits to prove global convergence.

- Our simulation results show that the new GAN performs better than WGAN-GP and the original GAN in standard datasets, in terms of FID scores. Our modification to the loss function has some similarity with the Wasserstein distance (coupling data points), but it turns out it works better than WGAN-GP on all datasets we tested. We remark that CP-GAN is orthogonal to other techniques such as spectral normalization Miyato et al. (2018), thus we only compare with vanilla methods such as WGAN-GP.

Finally, we remark that CP-GAN is just one example of GAN problems with nice global properties, and we hope our analysis can shed light on the design of other tractable GAN formulations.

**Outline**. The rest of the paper is structured as follows. We present the CP-GAN loss in Section 2, and discuss our analysis framework in Section 3. In Section 4, we analyze the outer-optimization problem of JS-GAN and the training dynamics of JS-GAN and W-GAN. In Section 5, we analyze the outer-optimization problem and the training dynamics of CP-GAN. We evaluate its performance on synthetic and standard datasets in Section 6. All proofs are in the Appendix.

## 2 New Loss: CP-GAN

In this section, we first review the formulation of GANs briefly. Next, we present the new formulation of CP-GAN. The theoretical advantage of the new formulation will be discussed later.

### 2.1 Generative Adversarial Networks

We first review the formulation of the original GAN proposed by Goodfellow et al. (2014). Given samples from a true data distribution $p_{\mathrm{data}}$, we want to generate a new distribution $p_{\mathrm{g}}$ to mimic $p_{\mathrm{data}}$. To judge how far away the generated distribution $p_{\mathrm{g}}$ is from $p_{\mathrm{data}}$, a discriminator (a.k.a. critic) computes a loss value that measures their gap. This discriminator $D$ addresses a binary classification problem, which should yield 1 for the true data point and 0 for the generated data point. The goal of the generator is to generate $p_{\mathrm{g}}$ to fool the discriminator so that the loss value is minimized. Formally, the problem for a GAN is given by

$$\min_{p_{\mathrm{g}}} \max_{D} E_{x \sim p_{\mathrm{data}}, y \sim p_{\mathrm{g}}} \log(D(x)) + \log(1 - D(y)). \tag{1}$$

A common choice for the discriminator is $D(u) = \frac{1}{1+\exp(-f(u))}$, where $f$ is a function. Given this choice, the formulation given in Eq. (1) becomes

$$\min_{p_{\mathrm{g}}} \max_{f} E_{x \sim p_{\mathrm{data}}, y \sim p_{\mathrm{g}}} \log \frac{1}{1 + \exp(-f(x))} + \log \frac{1}{1 + \exp(f(y))}.$$

We will use a generator $G_\theta(z)$ parameterized by $\theta$ to produce samples from a distribution $p_\text{g}$, where $z$ is drawn from a given distribution $p_z$. This yields the following program:

$$\min_G \max_f E_{x \sim p_\text{data}, z \sim p_z} \log \frac{1}{1 + \exp(-f(x))} + \log \frac{1}{1 + \exp(f(G(z)))}. \tag{2}$$

To resolve the vanishing gradient issue, Goodfellow et al. (2014) proposed to use a game formulation (often referred to as non-saturating GAN):

$$\max_f E_{x \sim p_\text{data}, z \sim p_z} \log \frac{1}{1 + \exp(-f(x))} + \log \frac{1}{1 + \exp(f(G(z)))}, \tag{3a}$$

$$\min_G E_{x \sim p_\text{data}, z \sim p_z} \log(1 + \exp(-f(G(z)))). \tag{3b}$$

## 2.2 COUPLING GAN

As mentioned before, the program is given in Eq. (3) has shortcomings, which we propose to address via the following formulation:

$$\min_\psi \max_\theta \mathbb{E}_{x \sim p_\text{data}, z \sim p_z}[\log \frac{1}{1 + \exp(f_\theta(G_\psi(z)) - f_\theta(x))}]. \tag{4a}$$

Similar to the $-\log D$ trick used in the original GAN training (a.k.a. non-saturating GAN), in practice we always use the non-saturating version for training:

$$D \text{ problem} : \min_\theta \mathbb{E}_{x \sim p_\text{data}, z \sim p_z}[\log(1 + \exp(f_\theta(G_\psi(z)) - f_\theta(x)))], \tag{5a}$$

$$G \text{ problem} : \min_\psi \mathbb{E}_{x \sim p_\text{data}, z \sim p_z}[\log(1 + \exp(f_\theta(x) - f_\theta(G_\psi(z))))]. \tag{5b}$$

Here $f_\theta$ is a deep net parameterized by $\theta$, and $G_\psi$ is a generator net parameterized by $\psi$.

How to interpret the new loss? Our original intuition is to use "personalized criterion". In JS-GAN, the discriminator wants to solve a binary classification problem by finding a single decision boundary that separates true data and generated data. For instance, if true data are $x_1 = 1$ and $x_2 = 2$, and generated data are $y_1 = 0.5$ and $y_2 = 0.5$, then the boundary is drawn at, say, $0.8$. Then the discriminator only has the motivation to approach $x_2 = 1$, since this would cross the bar of the current discriminator. An analogy is that if the requirement for the students is "get 60 points", then they will try to get 60 points and then rest (surely, there will be follow-up raise, but the raise cannot be too fast, since most students cannot catch). This causes $x_2$ to be missed by the generator, since it is "too good to learn". In contrast, in CP-GAN, the discriminator provides two boundaries at $0.8$ and $1.5$, and $y_1$ will try to cross $0.8$ to approach $x_1 = 1$, and $y_2$ will try to cross $1.5$ to approach $x_2 = 2$. Thus it can better learn the true distribution. An analogy is that if the requirements are set differently for different students, like one "60 point" and one "90 point", then they will learn to achieve the two goals. Thus CP-GAN is adopting "personalized criterion" for the generated data.

## 3 MULTI-DIRACGAN: MULTI-MODE GENERATION

Mescheder (2018) quoted Rahimi: "simple experiments, simple theorems are the building blocks that help us understand more complicated systems." Then they defined a simple model called Dirac-GAN, which uses a linear discriminator and a powerful generator to recover a single-point distribution. We define a much more general model than the DiracGAN. We refer it as Multi-DiracGAN.

**Definition 3.1** *The Multi-DiracGAN consists of a generator distribution $Y = (y_1, \ldots, y_n) \in \mathbb{R}^{d \times n}$ and a discriminator $f(x)$. The true data distribution is given by a $n$-point distribution $X = (x_1, \ldots, x_n) \in \mathbb{R}^{d \times n}$, where $x_i$'s are distinct.*

One motivation of the Multi-DiracGAN is to consider an "empirical" version of the problem. Yet another motivation is the construction of a simple model of multi-mode distributions: for instance, when $n = 2$, we have a two-point distribution which is the simplest two-mode distribution. We are not aware of an existing global analysis that can be applied to this simple 2-point model. See Appendix A for more discussions.

When analyzing an optimization formulation, it is natural to proceed in the following steps: (1) Sanity check, i.e., study whether the globally optimal solutions are desired; (2) Landscape analysis, i.e., check whether there are undesirable local minima; (3) Convergence analysis, i.e., check whether the proposed algorithm converges to a local minimum or stationary point. We will first perform a sanity check and landscape analysis (for powerful discriminators), and then study the dynamics (for linear discriminators).

## 4 ANALYSIS OF SOME EXISTING GANS

### 4.1 JS-GAN HAS EXPONENTIALLY MANY BAD BASINS

We analyze the landscape of JS-GAN under the Multi-DiracGAN model with powerful discriminators, the formulation of which can be written as:

$$\min_{Y \in \mathbb{R}^n} \phi_{\mathrm{JS}}(Y, X) \triangleq \max_f L^{\mathrm{JS}}(f; Y), \tag{6}$$

where $L^{\mathrm{JS}}(f; Y) = -\frac{1}{n} \sum_{i=1}^n \log(1 + \exp(-f(x_i))) - \frac{1}{n} \sum_{i=1}^n \log(1 + \exp(f(y_i)))$.

The range of $\phi_{\mathrm{JS}}(Y, X)$ is $[-2\log 2, 0]$ because $L^{\mathrm{JS}}(f; Y) \leq 0$ and $\phi_{\mathrm{JS}}(Y, X) \geq L(0; Y) = -2\log 2$, where $L(0; Y)$ represents the value achieved at $f = 0$.

To build intuition, we first present a result for the case of $n = 2$ points.

**Claim 4.1** *Suppose $n = 2$ and $x_1 \neq x_2 \in \mathbb{R}^d$. Then*

$$\phi_{\mathrm{JS}}(Y, X) = \begin{cases} -2\log 2 \approx -1.3862, & \text{if } \{x_1, x_2\} = \{y_1, y_2\} \\ -\log 2 \approx -0.6931, & \text{if } |\{x_1, x_2\} \cap \{y_1, y_2\}| = 1, \\ \log 2 - 1.5\log 3 \approx -0.9548, & \text{if } y_1 = y_2 \in \{x_1, x_2\}, \\ 0 & \text{if } |\{x_1, x_2\} \cap \{y_1, y_2\}| = \emptyset. \end{cases}$$

*The global minimum is $-2\log 2$, which is achieved iff the generated data points $\{y_1, y_2\}$ coincide with the true data points $\{x_1, x_2\}$.*

As a corollary of the above claim, the outer optimization objective of the original GAN has a bad strict local-min when two points overlap (a mode-collapse).

**Corollary 4.1** *Suppose $n = 2$ and $x_1 \neq x_2 \in \mathbb{R}^d$. Then $\bar{Y} = (x_1, x_1)$ is a sub-optimal strict local minimum of the function $g(Y) = \phi_{\mathrm{JS}}(Y, X)$.*

We illustrate the landscape of $\phi_{\mathrm{JS}}(Y, X)$ in Figure 1a for the special case that $d = 1, x_1 = 0, x_2 = 1$. In this case, $Y = (y_1, y_2) = (0, 1)$ is the global minimum of $\phi_{\mathrm{JS}}(Y, X)$ with optimal value $-2\log 2 \approx -1.3862$, and $Y = (0, 0)$ is a strict local minimum of $\phi_{\mathrm{JS}}(Y, X)$ with value approximately $-0.9548$. An intuitive way to understand the landscape and Corollary 4.1 is given as follows. Consider a point $(y_1, y_2)$ moving from $(0, 1)$ to $(-0.1, -0.1)$, going through four points: $(y_1, y_2) = (0, 1), (0, 0.5), (0, 0)$, and finally $(-0.1, -0.1)$. The four points correspond to the four cases above, thus giving values $-1.3862, -0.6931, -0.9548, 0$. The four values are not monotone, indicating that there is a strict local minimum at $-0.9538$.

For general $n$, the landscape can be characterized in a similar fashion. The following result states that the original GAN objective has exponentially many strict bad local minima.

**Proposition 1** *Suppose $x_1, x_2, \ldots, x_n \in \mathbb{R}^d$ are distinct. Consider the problem in Eq. (6).*

*(i) The global minimal value is $-2\log 2$, which is achieved iff the generated data points coincide with the true data points, i.e., $\{y_1, \ldots, y_n\} = \{x_1, \ldots, x_n\}$.*

*(ii) If $y_i \in \{x_1, \ldots, x_n\}, i = 1, 2, \ldots, n$ and $y_i = y_j$ for some $i \neq j$, then $Y$ is a sub-optimal strict local minimum. Therefore, $\phi_{\mathrm{JS}}(\cdot, X)$ has $(n^n - n!)$ sub-optimal strict local minima.*

The landscape analysis is a high-level analysis that provides some preliminary insight into the training process. But there are two gaps compared to the true training process: (1) in practice, we always

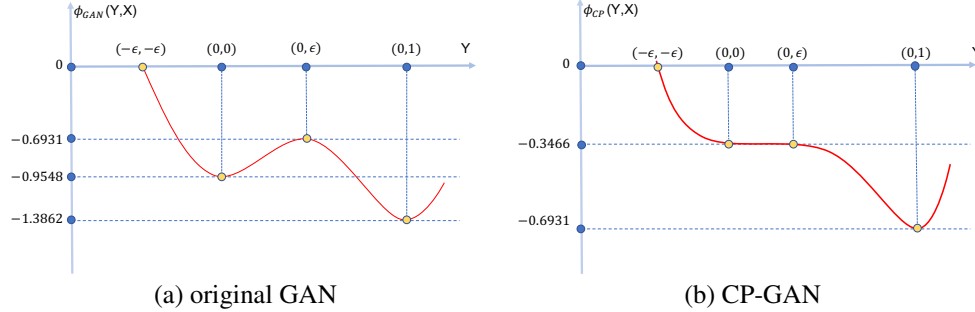

(a) original GAN  (b) CP-GAN

Figure 1: Graphical illustration of the landscape of GAN outer optimization problem $\min_Y \phi(Y, X)$. It is not a rigorous figure for two reasons: (1) there are only four possible function values, thus the function is a piece-wise linear function, but we use smooth curves to connect them to make the figure easier to understand. (2) the landscape should be two-dimensional, but we only pick illustrative points and and illustrate them in 1D space.

use different objectives for $D$ and $G$ (the $-\log D$ trick) to resolve the gradient vanishing issue; (2) the $D$ problem is addressed via a few gradient steps only. We suspect that the major insight conveyed in the landscape analysis can still carry over to the training dynamics for the following reasons: (1) the objectives for $D$ and $G$ are still closely related thus the "landscape" is still similar (this can be proved rigorously by considering a bi-level optimization problem, which we skip here); (2) inexactly optimizing $D$ can smooth the landscape, so that the flat regions in $\phi^{\mathrm{JS}}(\cdot, X)$ becomes smooth (this may be why having a discontinuous outer-function is not too big an issue of JS-GAN). However, we suspect that this smoothing effect cannot easily eliminate some deep basins, especially when there are exponentially many. A complete analysis of this effect is beyond the scope of this paper, and we will just show some evidence by analyzing the training dynamics of JS-GAN with linear discriminators.

## 4.2 TRAINING DYNAMICS OF JS-GAN

We consider the non-saturating version of the formulation in Eq. (6), which is also the finite-sample version of Eq. (3). We assume a linear discriminator $f(u) = w^T u + b$, where $w \in \mathbb{R}^d$, $b \in \mathbb{R}$.

$$\min_{\theta=(w,b)\in\mathbb{R}^d\times\mathbb{R}} L^D_{\mathrm{GAN}}(Y;\theta) \triangleq \sum_{i=1}^n \log(1+\exp(-w^T x_i - b)) + \log(1+\exp(w^T y_i + b)),$$

$$\min_{Y=(y_1,\ldots,y_n)\in\mathbb{R}^{d\times n}} L^G_{\mathrm{GAN}}(Y;\theta) \triangleq \sum_{i=1}^n \log(1+\exp(-w^T y_i - b)).$$

Consider the dynamics corresponding to the simultaneous gradient descent (GD) [1]:

$$\frac{dw}{dt} = -\frac{\partial L^D_{\mathrm{GAN}}}{\partial w} = \sum_{i=1}^n \frac{x_i}{1+e^{w^T x_i + b}} - \frac{y_i}{1+e^{-w^T y_i - b}}, \tag{7a}$$

$$\frac{db}{dt} = -\frac{\partial L^D_{\mathrm{GAN}}}{\partial b} = \sum_{i=1}^n \frac{1}{1+e^{w^T x_i + b}} - \frac{1}{1+e^{-w^T y_i - b}}, \tag{7b}$$

$$\frac{dy_i}{dt} = -\frac{\partial L^G_{\mathrm{GAN}}}{\partial y_i} = \frac{w}{1+e^{w^T y_i + b}}. \tag{7c}$$

The common method to deal with such complicated dynamics is local linearization. This permits to analyze the local behavior of the dynamics (see, e.g., Mescheder et al. (2018)). A global analysis of the dynamics often requires a proper Lyapunov function that is non-increasing along the trajectories. But the design of Lyapunov functions is often challenging.

---

[1] Other papers, such as Mescheder (2018) analyzed both simultaneous GD and alternating GD. For simplicity, we only analyze simultaneous GD in this paper.

Even if one does not find such a function, one may wonder whether this is due to the lack of technical tools, or due to some intrinsic barriers. Motivated by the landscape result in Proposition 1, we wish to analyze the mode-collapse patterns, i.e., the pattern occurring when some points overlap. Interestingly, we find that collapsed modes cannot be recovered under JS-GAN dynamics, as formally stated in the following claim:

**Claim 4.2** *Starting from any initial point* $(\theta(0), Y(0))$, *the trajectory* $(\theta(t), Y(t))$ *defined in Eq.* (20) *satisfies* $\|y_i(t) - y_j(t)\| \leq \|y_i(0) - y_j(0)\|, \forall t$.

This claim predicts that starting from $y_1(0) = y_2(0) = \cdots = y_n(0)$ (all points fall into one mode), prevents these points from separating. It is interesting to more rigorously characterize bad global behavior of JS-GAN dynamics, either for a linear or a neural-net discriminator (e.g., whether mode-collapse patterns create bad attractors or limit cycles). Anyhow, the above claim makes it very hard, if not impossible, to prove the global convergence of JS-GAN dynamics for multi-mode problems.

### 4.3 Discussions of WGAN and its variants

One may wonder whether W-GAN or its variants can provide a solution to the above issues. We briefly discuss the difficulties in a global analysis of WGAN and its variants.

For W-GAN with Lipschitz constrain on $f$, it is not hard to prove that the outer-optimization problem has a nice landscape. However, it is hard to impose the Lipschitz condition in practice; in addition, gradient dynamics with constraints are difficult to analyze [2].

A practical solution to resolve the Lipschitz constraint issue is to add a gradient penalty, which recovers WGAN-GP Gulrajani et al. (2017). However, the outer-optimization problem $\min_Y \max_f \sum_i f(x_i) - \sum_i f(y_i) - \lambda \sum_i (\|\nabla_u f(u)\|_{u=y_i}\| - 1)^2$ may have a complicated landscape due to the extra regularization term.

We further show that the dynamics of WGAN-GP with linear discriminators have a similar issue to JS-GAN. The details are provided in Appendix.

**Claim 4.3** *Starting from any initial point* $(w(0), Y(0))$, *the trajectory* $(w(t), Y(t))$ *defined by WGAN-GP dynamics satisfies* $\|y_i(t) - y_j(t)\| = \|y_i(0) - y_j(0)\|, \forall t$.

To avoid additional issues due to constraints, we remain interested in the logistic function used in the JS-GAN formulation. We hope that a small modification to the JS-GAN directly addresses its issues. We will now show that CP-GAN does achieve this goal.

## 5 Analysis of CP-GAN

In this section, we analyze the proposed new CP-GAN formulation and show that it has an advantage over the original GAN framework. We highlight again a major difference of our analysis compared to earlier works: we focus on global analysis while most existing results focus on local analysis.

### 5.1 CP-GAN has No Bad Basin

Following the aforementioned steps, the finite-sample version of CP-GAN reads

$$\min_Y \phi_{\text{CP}}(Y, X), \text{ where } \phi_{\text{CP}}(Y, X) \triangleq \sup_f \frac{1}{n} \sum_{i=1}^{n} \log \frac{1}{1 + \exp(f(y_i) - f(x_i)))}, \qquad (8)$$

resulting in a range $\phi_{\text{CP}}(Y, X) \in [-\log 2, 0]$. For simplicity, we define $g^{\text{CP}}(Y) = \phi_{\text{CP}}(Y, X)$ as the data $X$ are fixed throughout the paper.

**Proposition 2** *Suppose* $x_1, x_2, \ldots, x_n \in \mathbb{R}^d$ *are distinct. The global minimal value of* $g^{\text{CP}}(Y)$ *is* $-\log 2$, *which is achieved iff* $\{x_1, \ldots, x_n\} = \{y_1, \ldots, y_n\}$. *Furthermore, for any* $Y$, *there is a continuous path from* $Y$ *to a global minimum along which the value of* $g^{\text{CP}}(Y)$ *is non-increasing.*

---

[2]Mescheder (2018) analyzed W-GAN dynamics, but only consider the local region around 0, thus its analysis essentially ignores the Lipschitz constraint.

To understand this result, consider the special case $n = 2$ and $x_1 \neq x_2 \in \mathbb{R}^d$. In this case, we have

$$\phi_{\text{CP}}(Y, X) = \begin{cases} -\log 2 \approx -0.6931, & \text{if } \{x_1, x_2\} = \{y_1, y_2\} \\ -\frac{1}{2}\log 2 \approx -0.3466, & \text{if } |\{i : x_i = y_i\}| = 1 \\ 0 & \text{otherwise.} \end{cases}$$

We illustrate the landscape of $g^{\text{CP}}(Y)$ in Figure 1b for the special case that $n = 2, d = 1, x_1 = 0, x_2 = 1$. In this case, $(y_1, y_2) = (0, 1)$ and $(y_1, y_2) = (1, 0)$ are two global minima of $g^{\text{CP}}(Y)$ with optimal value $-\log 2 \approx -0.6931$. Similar to the previous subsection, there is an intuitive way to understand the landscape: Consider a point $(y_1, y_2)$ moving from $(0, 1)$ to $(-0.1, -0.1)$, going through four points: $(y_1, y_2) = (0, 1)$, then $(0, 0.5)$, then $(0, 0)$, and finally $(-0.1, -0.1)$. The four points correspond to four values $-0.6931, -0.3466, -0.3466, 0$. The four values are non-decreasing, indicating that there is a monotone path connecting the mode-collapsed pattern $(0, 0)$ and a global-min $(0, 1)$. This is different from the landscape of the JS-GAN loss, where any path between the two points has to cross some barrier.

Comparing Proposition 2 and Proposition 1, we observe the landscape of CP-GAN to be better than that of the original GAN formulation. How does that help training? Intuitively, the original GAN landscape has many basins of attraction. When starting a random initial point, a descent algorithm might fall into one of the bad basins, causing mode collapse. For CP-GAN, the only basin of attraction is the global minimum, thus the algorithm is more likely to converge to the global minimum.

## 5.2 Training Dynamics of CP-GAN

Next, we consider the training dynamics of CP-GAN and reveal nice global properties. Consider the game variant of the GAN formulation in Eq. (8), i.e., the finite-sample version of Eq. (5). Further, we assume the linear discriminator $f(u) = w^T u$, where $w \in \mathbb{R}^d$:

$$\min_{\theta = w \in \mathbb{R}^d} L_{\text{CP}}^D(Y; \theta) \triangleq \sum_{i=1}^{n} \log(1 + \exp(w^T(y_i - x_i))),$$

$$\min_{Y = (y_1, \ldots, y_n) \in \mathbb{R}^{d \times n}} L_{\text{CP}}^G(Y; \theta) \triangleq \sum_{i=1}^{n} \log(1 + \exp(w^T(x_i - y_i))). \tag{9}$$

Consider the dynamics corresponding to the simultaneous gradient descent for CP-GAN:

$$\frac{dw}{dt} = -\frac{\partial L_{\text{CP}}^D}{\partial w} = \sum_{i=1}^{n} \frac{x_i - y_i}{1 + e^{w^T(x_i - y_i)}}, \tag{10a}$$

$$\frac{dy_i}{dt} = -\frac{\partial L_{\text{CP}}^G}{\partial y_i} = \frac{w}{1 + e^{w^T(y_i - x_i)}}. \tag{10b}$$

Define $\delta_i = y_i - x_i, \forall i$. Define function $\beta_{ij}(t) = \frac{1}{2}\|\delta_i(t) - \delta_j(t)\|^2, \forall i, j$.

**Claim 5.1** *Along the trajectory defined by Eq. (10), we have* $\|(\delta_i(t) - \delta_j(t)\| \leq \|(\delta_i(0) - \delta_j(0)\|, \forall i, j.$

This claim looks similar to the result for the JS-GAN dynamics given in Claim 4.2. However, there is an important difference. For the original GAN, the gaps between *generated samples* are shrinking, while for CP-GAN, the gaps between the *errors* are shrinking. Thus CP-GAN is a more natural choice from this perspective.

This claim only shows the difference of the two GAN formulations, and we will show that CP-GAN has the extra benefit that it has a natural potential function. Consider the function

$$V = \frac{1}{2}\|w\|^2 + \frac{1}{2}\sum_{i=1}^{n}\|x_i - y_i\|^2.$$

Obviously, $V = 0$ iff $w = 0$ and $x_i = y_i, \forall i$. According to the following result, $V$ is non-increasing along the trajectory of CP-GAN training.

| Model | MNIST | | CIFAR10 | | STL10 | | CelebA | | LSUN | |
|---|---|---|---|---|---|---|---|---|---|---|
| | IS | FID | IS | FID | IS | FID | IS | FID | IS | FID |
| NSGAN | 1.98±0.01 | 7.73 | 5.62±0.09 | 53.42 | 5.88±0.06 | 82.81 | 2.46±0.01 | 21.68 | 2.77±0.03 | 41.15 |
| WGAN-GP | 2.00±0.01 | 7.05 | 6.52±0.08 | 40.31 | 7.34±0.11 | 65.23 | 2.65±0.01 | 17.18 | 2.80±0.01 | 34.02 |
| CPGAN | 2.05±0.02 | 5.08 | 6.72±0.06 | 36.66 | 7.56±0.21 | 57.07 | 2.73±0.02 | 15.45 | 2.82±0.02 | 30.41 |

Table 1: Inception score (IS) (higher is better) and Frechét Inception distance (FID) (lower is better) for non-saturating GAN, WGAN-GP and our proposed CPGAN on MNIST, CIFAR10, STL10, CelebA and LSUN.

**Claim 5.2** *Along the trajectory of the dynamics defined in Eq.* (10)*, we have* $\frac{dV(t)}{dt} \leq 0$.

Thus $V$ is a Lyapunov function for the CP-GAN dynamics. As mentioned earlier Mescheder (2018) showed that $V$ is a Lyapunov function for the JS-GAN dynamics (non-saturating version) given in Eq. (20) if $n = 1$ and $x_1 = 0$. However, when we check the case $x_1 \neq 0$, we found it is no longer a Lyapunov function. This means that the JS-GAN formulation is not "translation-invariant" in the sense that shifting $x_i$ and $y_i$ together leads to different dynamics. A simple fix to make it shift-invariant is to a couple $Y$ and $X$ together.

With the Lyapunov function, we have the global convergence to the set $\theta$ such that $\langle \nabla_\theta V(\theta), h(\theta) \rangle = 0$, where $h$ is the right-hand side of Eq. (10). With a bit more effort, we can show the global convergence of CP-GAN dynamics.

**Proposition 3** *(global convergence of CP-GAN dynamics) Starting from any initial point $\theta(0)$, consider the trajectory defined by the CP-GAN dynamics $\theta(t)$.*

*(1) For general d,* $\lim_{t\to\infty} \theta(t) \in E$, *where* $E \triangleq \{(w, y) : w^T(y_i - x_i) = 0, \forall i\}$.

*(2) When d = 1, starting from any initial point $\theta(0)$,* $\lim_{t\to\infty} \theta(t) \in M$, *where* $M = \{(w, y) : w = 0, \sum_i x_i = \sum_i y_i\}$.

The set of stationary points include some undesired points. The next result shows that the Jacobian at these undesired points has a positive real part, thus they are not stable. Simulations also show that small perturbation can escape these undesired points.

**Claim 5.3** *When d = 1, for the training dynamics given in Eq.* (10)*, at any point other than the desired solution* $(y_1, \ldots, y_n) = (x_1, \ldots, x_n)$ *in the set of stationary solutions, the Jacobian has an eigenvalue with positive real part.*

Now we have a clear understanding of the global behavior of the training dynamics of CP-GAN. The most important fact is that the energy function $V$ is a Lyapunov function. There may be other loss functions that have a Lyapunov function, at least in the simple setting, and we leave that exploration to future work.

# 6 EXPERIMENTS

In this section, we present empirical results comparing different GAN loss functions. We aim to show the qualitative and quantitative improvements achieved by using the proposed loss.

## 6.1 SYNTHETIC DATASETS

We first demonstrate our results on 2-dimensional synthetic data. Specifically, the 2-dimensional data $x = (x_1, x_2)$ is drawn from a mixture of 5 or 25 equally weighted Gaussians each with a variance of 0.002, the means of which are spaced equally on the unit circle. See the blue points in columns (a) and (d) of figure 2 for an illustration. Our generated samples are shown as the red points in (a) and (c) in figure 2 and we can see our generators can catch every mode. The generator and the discriminator loss are shown in (b), (c), (e) and (f).

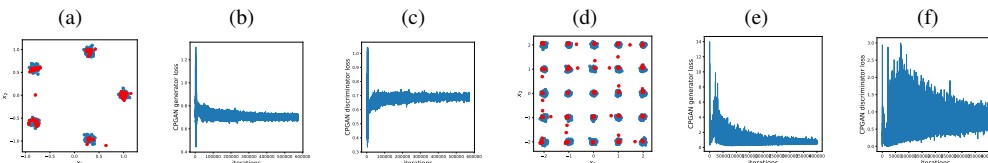

Figure 2: (a) True(blue) and generated(red) Samples (b) loss for the generator (c) loss for the discriminator from 5-Gaussian synthetic data. (d) True(blue) and generated(red) Samples (e) loss for the generator (f) loss for the discriminator from 25-Gaussian synthetic data.

## 6.2 IMAGE GENERATION

In this section, we present results on the task of image generation. We train and evaluate CPGAN on four datasets: (1) MNIST (LeCun et al., 1998) (32×32); (2) CIFAR-10 (Krizhevsky, 2009) (32×32); (3) STL-10 (Adam Coates, 2011) (48×48); (4) CelebA (Liu et al., 2015) (64×64); and (5) LSUN Bedrooms (Yu et al., 2015) (64×64). We use three ResNet Blocks structure for MNIST generator, and the DCGAN structure for CIFAR-10, STL-10, CelebA and LSUN. The input size for the generator is 128-dimension.

In all our experiments we train the generator and discriminator in an alternating fashion. The batch size is set to $64$. We trained the generator and the discriminator each $30k$ batch iterations on the MNIST dataset and $100k$ iterations on the CIFAR-10, STL-10, CelebA and LSUN datasets. We tuned the learning rate for each model to achieve their best performance. Some generated samples are shown in Appendix H.

We use inception scores (IS) (Salimans et al., 2016) and Frechét Inception distance (FID) (Heusel et al., 2017) to assess the quality of the generated images. We used $50k$ generated images for IS and $10k$ true images and $10k$ generated images for FID metric calculation. We provide the results in Table 1. We observe the proposed loss to perform better than other losses in terms of Inception Score and FID, on all data sets.

## 7 CONCLUSION

The analysis of the population version of GANs is difficult. In this work, we analyzed an empirical version of a few GAN formulations, i.e., assuming the true data distribution is the empirical distribution consists of $n$ points. In this formulation, the samples are moving during the optimization process, which is what is happening in practical training. This analysis can also mimics an $n$-mode distribution; in fact, it captures the macro-learning part of the learning process. We show that using this perspective, the JS-GAN formulation has exponentially many bad basins, and they correspond to mode collapse. We also show that a new formulation CP-GAN does not have bad basins. Further, we analyzed the training dynamics for the log-linear discriminators, and showed that there is a global Lyapunov function for CP-GAN. Simulation on CELEBA, CIFAR10, etc. shows that the new loss improves FID scores compared to WGAN-GP. This is achieved by only changing two lines of code in Pytorch, which we think is quite remarkable.

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

## A   EMPIRICAL VERSION: EFFECT OF MACRO-LEARNING

The motivation of analyzing the empirical version is two-fold. First, it is a common practice in machine learning to separately analyze the population version and the empirical version. Second, as we mentioned, "the $n$-point distribution is an estimation of the $n$-mode distribution", and we elaborate below.

GANs are implicit generative models, as opposed to the explicit models like energy based models (LeCun et al. (2006)) and restricted boltzman machine (Salakhutdinov & Hinton (2009)). In the implicit models, there is no explicit parameterization of the probabality density. The benefit of implicit models is that it is easier to train and sample from. During the training of GANs, when the parameters of the mapping (i.e. generator) change, it is the generated samples that are moving, not that an explicit probability density is moving. These two views are illustrated in Figure 3.

Next, we will explain that the two views are both related and orthogonal.

First, there is a connection between the two views. As shown in Figure 4, we can smooth the empirical distribution (blue color) which is discrete to be a continuous distribution (yellow color). During the algorithms, the samples are actually moving, but that also corresponds to the moving of the underlying continuous probability density. Thus it is reasonable to assume $p_\mathrm{g}$ is moving, as an approximation to the real picture that the samples are moving.

Second, we think these two views are orthogonal, and correspond to "macro-learning" and "micro-learning" respectively, when learning a multi-mode distribution. Consider learning a two-mode

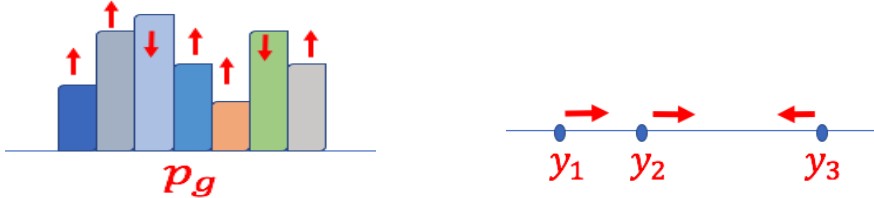

(a) Population version: probability density changes    (b) Empirical version: samples move

Figure 3: Population version and empirical version. Population version: the probability densities are changing. Empirical version: the samples are moving.

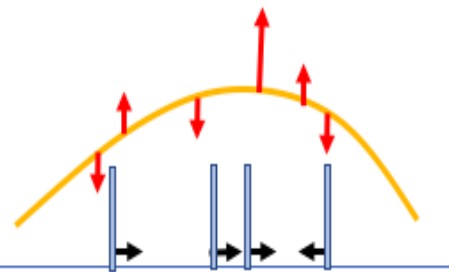

Figure 4: Illustration of the learning process of the single mode. The generated samples are moving, which correspond to the adjustment of the probability densities. They represent two views on the learning process.

distribution $p_{\text{data}}$, and we start from an initial two-mode distribution $p_{\text{g}}$. There are two differences between $p_{\text{g}}$ and $p_{\text{data}}$: first, the locations of the two modes are different; second, the distribution within each mode is different. To learn the distribution, we want to eliminate both differences: first, move the two modes of $p_{\text{g}}$ to roughly overlap with the two modes of $p_{\text{data}}$ which we call "macro learning"; second, adjust the distributions of each mode to match those of $p_{\text{data}}$, which we call "micro learning". This is illustrated in Figure 5 and Figure 6.

Therefore, we have explained the two reasons of analyzing the empirical version (the $n$-point distribution). First, it is the practically used version. Second, it captures the "macro-learning" behavior of learning a multi-mode distribution.

Finally, the two types of learning are related to the two types of mode collapse mentioned in Lin et al. (2018): "entire modes from the input data are never generated, or the generator only creates images within a subset of a particular mode." Failure of macro-learning can cause missing modes in the generated distributions, which corresponds to the first type of mode collapse. Failure of micro-learning can cause a sub-mode of a mode to be missed, which also corresponds to the second type of mode collapse. See more discussions of related works on mode collapse in Appendix B.

## A.1 GENERALIZATION

One may wonder whether fitting the $n$ data points can cause memorization and thus may not generalize. We explain from two perspectives: first, the existing theory on generalization of GANs; second, intuition why this does not overfit.

First, there exists generalization bounds for GANs. Suppose the true distribution is $p_{\text{data}}$ and the generated distribution is $p_{\text{g}}$. Further, suppose we sample $n$ points $x_i's$ from $p_{\text{g}}$ and sample $n$ points from $p_{\text{g}}$, and use $\mu$ and $\nu$ to represent the two empirical distributions (discrete distributions with equal probability). Arora et al. (2017) proved that for a large class of GAN problems (including JS-GAN using neural-net discriminator and generators), only polynomial samples are needed to achieve a small generalization error. As a result of the generalization bound, Arora et al. (2017)

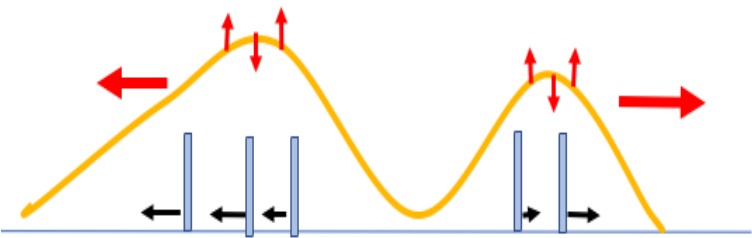

Figure 5: Illustration of the process of learning a multi-mode distribution. Will decompose this process into two parts in the next figure.

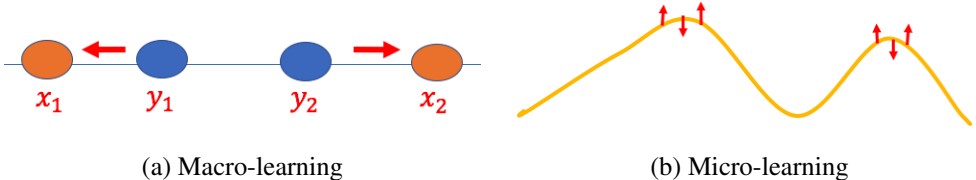

(a) Macro-learning           (b) Micro-learning

Figure 6: Illustration of learning a multi-mode distribution. We decompose the process as the macro-learning and the micro-learning. The macro-learning refers to the moving of the whole mode towards the underlying mode. The micro-learning refers to the adjustment of the distribution within each mode. If macro-learning fails, then an entire mode is missed in the generated distributions, which corresponds to mode collapse. Note that even for the single-mode, the learning process can be decomposed into the macro learning and the micro learning.

stated that if the GAN successfully minimized the empirical distance (i.e. the distance between $\mu$ and $\nu$), then the population distance (the distance between the two distributions $p_{\text{data}}$ and $p_{\text{g}}$) is also small. Therefore, there is a generalization guarantee under suitable conditions. Of course, there is much space in improving the generalization bound of Arora et al. (2017), and that is an orthogonal line of research. Our goal of this paper is mainly to study how to "successfully minimized the empirical distance" (i.e. fit the $n$-point distribution $\mu$). Technically speaking, the form of Arora et al. (2017) does not cover CP-GAN, but the proof can be easily extended to CP-GAN.

Second, we provide some intuition why fitting the points may not cause overfitting. Consider a simple case of learning a two-mode distribution in Figure 7. During training, we learned a generator that maps the latent samples $z_i$'s to $x_i$'s, thus fit the empirical distribution. If we sample a new latent sample $z_i$, then the generator will map $z_j$ to a new point $x_j$ in the underlying data distribution. Thus fitting the empirical distribution can still leads to generating new data points.

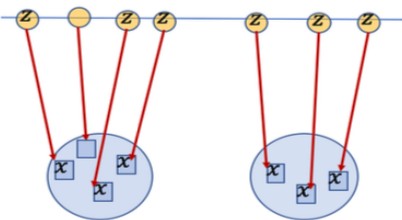

Figure 7: Learning a two-mode distribution. During training, we learned a generator that maps the latent samples $z_i$'s to $x_i$'s, thus fit the data. If we sample a new $z_i$, then it will be mapped to a new point in the underlying data distribution.

## B  RELATED WORKS

**Single-mode analysis.** Two papers Feizi et al. (2017) Mescheder et al. (2018) provided global analysis of GANs for the single-mode case. Feizi et al. (2017) analyzed the case that the true distribution $p_{\text{data}}$ is a single Gaussian distribution. It considers the population version, i.e., there is a continuous latent distribution of $p_z$. For a new formulation called quadratic GAN and using a linear generator and a quadratic discriminator, this paper proved the global convergence of alternating GDA. The min-max problem they consider is a bi-linear function for $D$ and a bi-linear function for $G$ (not a typical a bi-linear game which is linear in either variable, but bi-linear in either variable). They found an interesting Lyapunov function for the min-max problem, leading to the global convergence. The major difference with our papers is that they considered the single-mode case, while we consider the multi-mode case. There are a few other differences: (a) They consider the population version, and we consider the empirical version. (b) They consider the quadratic discriminator (since to learn a Gaussian, the disciminator needs to be constrained), and we analyze both the powerful discriminator case and the linear discriminator.

To extend to the multi-mode case such as multi-Gaussian, as we discussed earlier, there is a macro-learning effect and micro-learning effect. Our work on $n$-point distributions captures the macro-learning effect, and Feizi et al. (2017) captures the micro-learning effect for Gaussian data. In the future, it would be quite interesting to combine the analysis of Feizi et al. (2017) and our analysis to the multi-Gaussian case.

Mescheder et al. (2018) considered the case that the true distribution $p_{\text{data}}$ is a single point $0$. Even for this case, GDA for all min-max formulations of GANs does not converge locally. However, it proved that for the non-saturating version of JS-GAN, GDA converges to the desired solution $0$. Interestingly, we found that their proof cannot be directly generalized to the case that $p_{\text{data}}$ is a single non-zero point. Thus even for the single non-zero point setting, it is natural to use our CP-GAN formulation to prove global convergence.

**Mode collapse.** Mode collapse is one of the major challenges for GANs, and received a lot of attention. Roughly speaking, it means that some "modes" of the true data distributions are not generated. The cause of mode collapse is not well understood, and there are a few high-level hypotheses, such as improper loss function Arjovsky & Bottou (2017); Arora et al. (2017) and weak discriminators Metz et al. (2016); Salimans et al. (2016); Arora et al. (2017); Li et al. (2017). Interestingly, CP-GAN both changes the loss function and improves the discriminator.

A number of empirical solutions have been proposed, including unrolled GAN Metz et al. (2016) and minibatch discrimination Salimans et al. (2016). A recent work Lin et al. (2018) proposed a theoretically motivated method PacGAN, which we elaborate below. The key observation is the following: two pairs of distributions with the same total variation distance do not exhibit the same degree of mode collapse. For instance, consider $Q = U[0,1]$, $P_1 = U[0.2,1]$ and $P_2 = 0.6U[0,0.5] + 1.4U([0.5,1])$, where $U[a,b]$ is the uniform distribution on $[a,b]$. Then $TV(Q, P_1) = TV(Q, P_2)$, but $P_1$ has mode collapse while $P_2$ does not. They proposed to pack the samples, i.e., consider the distance of the product distribution $P^m$ and $Q^m$. The main theoretical result that "$TV(P^m, Q^m)$ is a better loss to penalize strong mode collapse than $TV(P, Q)$". Based on this result, they extracted the key idea of "packing" to apply it to any GAN.

Our paper is different from Lin et al. (2018) in the following aspects. First, they did not provide theoretical analysis for a specific GAN; in contrast, we prove theoretical results of specific JS-GAN and CP-GAN formulations. Second, we provided an explanation for "why mode collapse happens", by linking mode collapse to a fundamental optimization subject "bad basin". Third, their focus is to mitigate "bad basin", and our starting point is to analyze the global landscape, and the link to mode collapse is a natural byproduct of the analysis.

## C  PROOFS IN SECTION 4.1

### C.1  PROOF OF CLAIM 4.1

We will compute values of $\phi_{\text{JS}}(Y, X)$ for all $Y$.

Denote $D(u) = \frac{1}{1+\exp(-f(u))} \in [0,1]$. Since $f$ can be any continuous function, $D$ can be any continuous function with range $(0,1)$.

$$\phi_{JS}(Y,X) = \sup_f \left[ \frac{1}{n} \sum_{i=1}^n \log \frac{1}{1+\exp(-f(x_i))} + \frac{1}{n} \sum_{i=1}^n \log \frac{1}{1+\exp(f(y_i))} \right]$$

$$= \sup_D \left[ \frac{1}{n} \sum_{i=1}^n \log(D(x_i)) + \frac{1}{n} \sum_{i=1}^n \log(1 - D(y_i)) \right].$$

Consider four cases.

**Case 1**: Both generated points overlap with the true data points, and are distinct, i.e., $y_1 = x_1, y_2 = x_2$ or $y_2 = x_1, y_2 = x_2$. Then the objective is

$$\sup_D \left[ \frac{1}{2} \log(D(x_1)) + \frac{1}{2} \log(1 - D(x_1)) + \frac{1}{2} \log(D(x_2)) + \frac{1}{2} \log(1 - D(x_2)) \right].$$

The optimal value is $-2\log 2$, which is achieved when $D(x_1) = D(x_2) = \frac{1}{2}$. The corresponding function values of $f$ are $f(x_1) = f(x_2) = 0$. The values of $f$ on other points do not matter.

**Case 2**: Exactly one of the generated points overlaps with the true data points. Without loss of generality, we can check the case $y_1 = x_1, y_2 \notin \{x_1, x_2\}$. The problem becomes

$$\sup_D \left[ \frac{1}{2} \log(D(x_1)) + \frac{1}{2} \log(D(x_2)) + \frac{1}{2} \log(1 - D(x_1)) + \frac{1}{2} \log(1 - D(y_2)) \right].$$

The optimal value $-\log 2 \approx -0.6931$ is achieved when $D(x_1) = 1/2, D(x_2) = 1$ and $D(y_2) = 0$. The corresponding function values of $f$ are $f(x_1) = -\log 2, f(x_2) = \infty$ and $f(y_2) = -\infty$.

**Case 3**: Both generated points overlap with the true data points, and are the same. Without loss of generality, check the case $y_1 = y_2 = x_1$.

$$\sup_D \left[ \frac{1}{2} \log(D(x_1)) + \log(1 - D(x_1)) + \frac{1}{2} \log(D(x_2)) \right].$$

The optimal value $\frac{1}{2} \log \frac{1}{3} + \log \frac{2}{3} \approx -0.9548$ is achieved when $D(x_1) = 1/3$ and $D(x_2) = 0$. The corresponding function values of $f$ are $f(x_1) = -\log 2, f(x_2) = \infty$.

**Case 4**: Both generated points are different from the true data points. $y_1, y_2 \notin \{x_1, x_2\}$. It becomes separable over the four points

$$\sup_D \left[ \frac{1}{2} \log(D(x_1)) + \frac{1}{2} \log(D(x_2)) + \frac{1}{2} \log(1 - D(y_1)) + \frac{1}{2} \log(1 - D(y_2)) \right].$$

Each term can achieve its maximum $\log 1 = 0$. Thus the optimal value $0$ is achieved when $D(x_1) = D(x_2) = 1$ and $D(y_1) = D(y_2) = 0$. The corresponding function values of $f$ are $f(x_1) = -\infty$, $f(x_2) = \infty$.

## C.2 PROOF OF COROLLARY 4.1

We re-state the corollary below.

**Corollary C.1** *Suppose $\bar{Y} = (\bar{y}_1, \bar{y}_2)$ satisfies $\bar{y}_1 = \bar{y}_2 = x_1$, then it is a sub-optimal strict local minimum of the problem.*

**Proof**: Suppose $\epsilon$ is the minimal non-zero distance between two points of $x_1, x_2, y_1, y_2$. Consider a small perturbation of $\bar{Y}$ as $Y = (\bar{y}_1 + \epsilon_1, \bar{y}_2 + \epsilon_2)$, where $|\epsilon_i| < \epsilon$. We want to verify that

$$\phi(\bar{Y}, X) > \phi(Y, X) \approx -0.9548. \tag{11}$$

There are two possibilities.

**Possibility 1**: $\epsilon_1 = 0$ or $\epsilon_2 = 0$. WLOG, assume $\epsilon_1 = 0$, then we must have $\epsilon_2 > 0$. Then we still have $y_1 = \bar{y}_1 = x_1$. Since the perturbation amount is small enough, we have $y_2 \notin \{x_1, x_2\}$. According to Case 2 above, we have

$$\phi(\bar{Y}, X) = -\log 2 \approx -0.6931 > -0.9548.$$

**Possibility 2**: $\epsilon_1 > 0, \epsilon_2 > 0$. Since the perturbation amount $\epsilon_1$ and $\epsilon_2$ are small enough, we have $y_1 \notin \{x_1, x_2\}, y_2 \notin \{x_1, x_2\}$. According to Case 4 above, we have

$$\phi(\bar{Y}, X) = 0 > -0.9548.$$

Combining both cases, we have proved (12).

**Q.E.D.**

### C.3 PROOF OF PROPOSITION 1

Denote $D(u) = \frac{1}{1+\exp(-f(u))} \in [0, 1]$. Since $f$ can be any continuous function, $D$ can be any continuous function with range $(0, 1)$.

$$\phi_{\mathrm{JS}}(Y, X) = \sup_f \frac{1}{n} \sum_{i=1}^{n} \log \frac{1}{1 + \exp(-f(x_i))} + \frac{1}{n} \sum_{i=1}^{n} \log \frac{1}{1 + \exp(f(y_i))}$$

$$= \sup_D \frac{1}{n} \sum_{i=1}^{n} \log(D(x_i)) + \frac{1}{n} \sum_{i=1}^{n} \log(1 - D(y_i)).$$

Denote $F(D; Y) = \frac{1}{n} \sum_{i=1}^{n} \log(D(x_i)) + \frac{1}{n} \sum_{i=1}^{n} \log(1 - D(y_i))$. Since $D(u) \in (0, 1)$ for any $u$, then $F(D; Y) \leq 0$. When $D(u) \to \frac{1}{2}, \forall u$, we have $F(D; Y) \to -2 \log 2$, thus $\phi_{\mathrm{JS}}(Y, X) = \sup_D F(D; Y) \geq -2 \log 2$. Thus the range of $\phi_{\mathrm{JS}}(\cdot, \cdot)$ is $[-2 \log 2, 0]$.

Now we compute the value of $\phi_{\mathrm{JS}}(\cdot, X)$ for each $Y$. For any $i$, denote $M_i$ to be set of indices of $y_j$ such that $y_j$ equals $x_i$, and $m_i$ to be the size of the set $M_i$, i.e.,

$$M_i = \{j : y_j = x_i\}, m_i = |M_i|, i = 1, 2, \ldots, n.$$

Denote $\Omega = \{1, 2, \ldots, n\} \backslash (M_1 \cup M_2 \cdots \cup M_n)$. Then we have

$$\phi_{\mathrm{JS}}(Y, X) = \frac{1}{n} \sup_D \left( \sum_{i=1}^{n} [\log(D(x_i)) + \sum_{j \in M_i} \log(1 - D(y_j))] + \sum_{j \notin \Omega} \log(1 - D(y_j)) \right)$$

$$= \frac{1}{n} \sup_D \left( \sum_{i=1}^{n} [\log(D(x_i)) + m_i \log(1 - D(x_i))] + \sum_{j \notin \Omega} \log(1 - D(y_j)) \right)$$

$$\overset{(i)}{=} \frac{1}{n} \sum_{i=1}^{n} \sup_{t_i \in \mathbb{R}} [\log(t_i) + m_i \log(1 - t_i)] + 0$$

$$\overset{(ii)}{=} \frac{1}{n} \sum_{i=1}^{n} [\log \frac{1}{m_i + 1} + m_i \log \frac{m_i}{m_i + 1}]$$

$$= \frac{1}{n} \sum_{i=1}^{n} [m_i \log m_i - (m_i + 1) \log(m_i + 1)].$$

Here (i) is because $D(y_j), j \in \Omega$ are independent of $D(x_i)$'s and thus can be any values; in this step, the optimal $D(y_j) = 1, \forall j \in \Omega$. (ii) is because for any positive number $m$, $\sup_{t \in \mathbb{R}} [\log(t) + m \sum_{i=1}^{n} \log(1 - t)] = \log \frac{1}{m+1} + m \log \frac{m}{m+1}$.

When $m_i = 1, i = 1, 2, \ldots, n$, i.e., $y_i = x_i, \forall i$, the function $\phi_{\mathrm{JS}}(Y, X)$ achieves value $-2 \log 2$. Thus $Y = (x_1, x_2, \ldots, x_n)$ is a global minimum.

Next, we show that if $Y$ satisfies that if $m_1 + m_2 + \cdots + m_n = n$ then $Y$ is a strict local-min. Denote $\delta$ as the minimal distance between two points of $x_1, x_2, \ldots, x_n$, i.e.,

$$\delta = \min_{k \neq l} \|x_k - x_l\|.$$

Consider a small perturbation of $Y$ as $\bar{Y} = (\bar{y}_1, \bar{y}_2, \ldots, \bar{y}_n) = (y_1 + \epsilon_1, y_2 + \epsilon_2, \ldots, y_n + \epsilon_n)$, where $\|\epsilon_j\| < \delta, \forall j$ and $\sum_j \|\epsilon_j\|^2 > 0$. We want to verify that

$$\phi_{\mathrm{JS}}(\bar{Y}, X) > \phi_{\mathrm{JS}}(Y, X). \tag{12}$$

Denote
$$\bar{m}_i = |\{j : \bar{y}_j = x_i\}|, i = 1, 2, \ldots, n.$$
According to the assumption $m_1 + m_2 + \cdots + m_n = n$, we have $y_j \in \{x_1, \ldots, x_n\}, \forall j$. Consider an arbitrary $j$, and suppose $y_j = x_i$. Together with $\|\bar{y}_j - y_j\| = \|\epsilon_j\| < \delta = \min_{k \neq l} \|x_k - x_l\|$, we have $\bar{y}_j \notin (\{x_1, x_2, \ldots, x_n\}\backslash\{x_i\})$. In other words, the only possible point in $\{x_1, \ldots, x_n\}$ that can coincide with $\bar{y}_j$ is $x_i$, and this happens only when $\epsilon_j = 0$. As a result, we have
$$\bar{m}_i \leq m_i, \quad , i = 1, 2, \ldots, n.$$
Together with the fact that $t \log t - (t + 1) \log(t + 1)$ is a strictly decreasing function in $t \in [0, \infty)$, we have

$$\phi_{\text{JS}}(\bar{Y}, X) = \frac{1}{n} \sum_{i=1}^{n} [\bar{m}_i \log \bar{m}_i - (\bar{m}_i + 1) \log(\bar{m}_i + 1)]$$

$$\geq \frac{1}{n} \sum_{i=1}^{n} [m_i \log m_i - (m_i + 1) \log(m_i + 1)] = \phi_{\text{JS}}(Y, X).$$

The equality is achieved iff $\bar{m}_i = m_i, \forall i$, which holds iff $\epsilon_j = 0, \forall j$. Since we have assumed $\sum_j \|\epsilon_j\|^2 > 0$, the equality does not hold, thus $\phi(\bar{Y}, X) > \phi(Y, X)$. Therefore, we have proved that $\bar{Y}$ is a strict local minimum.

Finally, if $Y$ satisfies that $m_1 + m_2 + \cdots + m_n = n$ and $m_k \geq 2$ for some $k$, then $\phi_{\text{JS}}(Y, X) > -2 \log 2$. Thus $Y$ is a sub-optimal strict local minimum. Q.E.D.

## D  TECHNICAL DETAILS OF SECTION 4.2 AND 4.3

### D.1  PROOF OF CLAIM 4.2

Define a function $\alpha_{ij}(t) = \frac{1}{2}\|y_i(t) - y_j(t)\|^2, \forall i, j$.
$$\frac{d\alpha_{ij}(t)}{dt} = (y_i - y_j)^T \frac{d(y_i - y_j)}{dt} = (y_i - y_j)^T \left(\frac{dy_i}{dt} - \frac{dy_j}{dt}\right)$$

$$= (y_i - y_j)^T \left(\frac{w}{1 + e^{w^T y_i + b}} - \frac{w}{1 + e^{w^T y_j + b}}\right)$$

$$= (w^T y_i - w^T y_j) \left(\frac{1}{1 + e^{w^T y_i + b}} - \frac{1}{1 + e^{w^T y_j + b}}\right)$$

$$\leq 0.$$

In the last step, we used the fact that $(a_1 - a_2) \left(\frac{1}{1+e^{a_1}} - \frac{1}{1+e^{a_2}}\right) \leq 0$ for any $a_1, a_2 \in \mathbb{R}$. Note that in the above computation, we skip the time index $t$. Therefore $\alpha_{ij}(t)$ is non-increasing along the trajectory, which implies $\|y_i(t) - y_j(t)\| \leq \|y_i(0) - y_j(0)\|, \forall t$.

### D.2  PROOF OF CLAIM 4.3

WGAN-GP with linear discriminators can be expressed as

$$\min_{w \in \mathbb{R}^d} L^D_{\text{WGAN}-\text{GP}}(Y; \theta) \triangleq - \sum_{i=1}^{n} w^T x_i + \sum_{i=1}^{n} w^T y_i + \lambda(\|w\| - 1)^2.$$

$$\min_{Y=(y_1,\ldots,y_n) \in \mathbb{R}^{d \times n}} L^G_{\text{WGAN}-\text{GP}}(Y; \theta) \triangleq - \sum_{i=1}^{n} w^T y_i.$$

Consider the dynamics corresponding to the simultaneous gradient descent for WGAN-GP:

$$\frac{dw}{dt} = -\frac{\partial L^D_{\text{WGAN}-\text{GP}}}{\partial w} = \sum_{i=1}^{n} (x_i - y_i) - 2\lambda w + 2w/\|w\|, \tag{13a}$$

$$\frac{dy_i}{dt} = -\frac{\partial L^G_{\text{WGAN}-\text{GP}}}{\partial y_i} = -w. \tag{13b}$$

Define a function $\alpha_{ij}(t) = \frac{1}{2}\|y_i(t) - y_j(t)\|^2, \forall\, i, j$.

$$\frac{d\alpha_{ij}(t)}{dt} = (y_i - y_j)^T \frac{d(y_i - y_j)}{dt} = (y_i - y_j)^T \left( \frac{dy_i}{dt} - \frac{dy_j}{dt} \right)$$
$$= (y_i - y_j)^T (w - w)$$
$$= 0.$$

Thus $\|y_i(t) - y_j(t)\|$ will be a constant for all $t$.

## E  TECHNICAL DETAILS OF SECTION E.5

### E.1  PROOF OF CLAIM 5.1

Define a function $\alpha_{ij}(t) = \frac{1}{2}\|(y_i(t) - x_i) - (y_j(t) - x_j)\|^2, \forall\, i, j$. Denote $\delta_i(t) = y_i(t) - x_i$.

$$\frac{d\alpha_{ij}(t)}{dt} = (\delta_i - \delta_j)^T \frac{d(y_i - y_j)}{dt} = (\delta_i - \delta_j)^T \left( \frac{dy_i}{dt} - \frac{dy_j}{dt} \right)$$
$$= (\delta_i - \delta_j)^T \left( \frac{w}{1 + e^{w^T(y_i - x_i)}} - \frac{w}{1 + e^{w^T(y_j - x_j)}} \right)$$
$$= (w^T \delta_i - w^T \delta_j) \left( \frac{1}{1 + e^{w^T \delta_i}} - \frac{1}{1 + e^{w^T \delta_j}} \right)$$
$$\leq 0.$$

The equality holds iff $w^T \delta_i = w^T \delta_j$, i.e., $w^T(\delta_i - \delta_j) = 0$.

### E.2  PROOF OF CLAIM 5.2

Let $u_i = w^T(y_i - x_i), i = 1, \ldots, n$, we have

$$\frac{dV}{dt} = w^T \frac{dw}{dt} + \sum_i (y_i - x_i)^T \frac{dy_i}{dt} \tag{14a}$$

$$= w^T \sum_{i=1}^n \frac{x_i - y_i}{1 + e^{w^T(x_i - y_i)}} + \sum_i (y_i - x_i)^T \frac{w}{1 + e^{w^T(y_i - x_i)}} \tag{14b}$$

$$= \sum_{i=1}^n \frac{w^T(x_i - y_i)}{1 + e^{w^T(x_i - y_i)}} + \sum_{i=1}^n \frac{w^T(y_i - x_i)}{1 + e^{w^T(y_i - x_i)}} \tag{14c}$$

$$= \sum_{i=1}^n \frac{-u_i}{1 + e^{-u_i}} + \sum_{i=1}^n \frac{u_i}{1 + e^{u_i}} \tag{14d}$$

$$= \sum_{i=1}^n \sum_{i=1}^n \frac{u_i(1 - e^{u_i})}{1 + e^{u_i}} \leq 0, \tag{14e}$$

In the last step we used the fact that $u(1 - e^u) \leq 0, \forall u \in \mathbb{R}$. Note that in the above computation, we skip the time index $t$.

### E.3  PROOF OF PROPOSITION 3

We restate the proposition below.

**Proposition 4** *(restate; global convergence of CPGAN dynamics) When $d = 1$, starting from any initial point $\theta(0)$, the trajectory defined by the CPGAN dynamics $\theta(t)$ will satisfy $\lim_{t \to \infty} \theta(t) \in M$, where $M = \{(w, y) : w = 0, \sum_i x_i = \sum_i y_i\}$.*

We will use Lasalle's invariance principle (see, e.g. (Haddad & Chellaboina, 2011, Theorem 3.3) or (Cohen & Rouhling, 2017, Theorem 1)) to prove this result.

**Definition** (invariance set). A set $A$ is said to be invariant with respect to a differential equation $\frac{du}{dt} = h(u(t))$ if every solution to this equation starting in $A$ remains in $A$.

**Lemma 1** *(Lasalle's invariance principle) Consider a dynamical system defined by $\frac{du}{dt} = h(u(t))$, where $h$ is a vector mapping. Assume $h$ has continuous first partial derivatives and $h(0) = 0$. Let $K$ be an invariant compact set. Suppose there is a scalar function $V$ which has continuous first partial derivatives in $K$ and is such that $\tilde{V}(p) = \langle \nabla V(p), h(p) \rangle \leq 0, \forall p \in K$. Let $E$ be the set of all points $p \in K$ such that $\tilde{V}(p) = 0$. Let $M$ be the largest invariant set in $E$. Then for every solution $t$ starting in $K$, $u(t) \to M$ as $t \to \infty$.*

Our goal is to show that starting from any initial point $\theta(0) = (w(0), y(0))$, the dynamics defined by (10), denoted as $\theta(t) = (w(t), y(t))$ will converge to the set

$$M = \{(w, y) : w = 0, \sum_i x_i = \sum_i y_i\}. \tag{15}$$

Define

$$K = \{\theta : \|\theta\|^2 \leq \|\theta(0)\|^2\}$$

Since $\dot{V} \leq 0$, we have $\|\theta(t)\|^2 \leq \|\theta(0)\|^2$, thus $\theta(t) \in K, \forall t$. Thus $K$ is an invariant compact set.

According to Claim 5.2, the set of points $p$ such that $\tilde{V}(p) = 0$ is

$$E \triangleq \{(w, y) : w^T (y_i - x_i) = 0, \forall i\}.$$

According to the LaSalle's invariance principle, the algorithm will converge to the largest invariance subset of $E$. Without knowing which set it is, we can already obtain the result that the algorithm converges to the set $E$. This proves the first part of the result. The analysis so far works for any $d$.

Next, we will use the condition $d = 1$. We show that when $M$ defined in (15) is the largest invariant set in $E$. Assume the contrary, that there is a set $M \subsetneq \bar{M} \subseteq E$ such that $\bar{M}$ is an invariant set. Then starting from any point $\theta(0) \in E \backslash M$, we have $\theta(t) \in \bar{M}$.

Since $d = 1$, then set $E$ becomes

$$E = \{(w, y) \in \mathbb{R}^{1 \times (n+1)} : w(y_i - x_i) = 0, \forall i\}.$$

Let $E_1 = \{(w, y) : w = 0\}$ and $E_2 = \{(w, y) : y_i = x_i, \forall i\}$, then $E = E_1 \cup E_2$. The set $M$ is still

$$M = \{(w, y) : w = 0, \sum_i x_i = \sum_i y_i\}.$$

Since $\theta(0) \in E$ but not in $M$, and $E_2 \subseteq M$, then we have $\theta(0) \in E_1$ and $\theta(0) \notin E_2$. This implies $w(0) = 0$ and $\sum_i (x_i(0) - y_i(0))^2 \neq 0$. According to the fact that $\tilde{V}(p) = 0, \forall p \in E$ and $\theta(t) \in E$, we have that

$$V(\theta(t)) = V(\theta(0)) = \sum_i (x_i(0) - y_i(0))^2 + w(0)^2 \triangleq \delta. \tag{16}$$

Since the path $\theta(t) \in M \subseteq E$ for all $t$, we have $\theta(t) \in E_1$ or $E_2$. Suppose $J = \{t \in \mathbb{R} : \theta(t) \in E_2\}$, then for $s \in J$ we have $V(\theta(s)) = w(s)^2 + 0 = w(s)^2$. By (16), we have

$$w(s)^2 = \delta, \forall s.$$

For $t \notin J$, we have $\theta(t) \in E_1$ and thus $w(t) = 0$. Thus $w(t) = 0, \forall t \notin J$ and $w(t) = \delta, \forall s \in J$. As $w(t)$ is continuous for $t$ and $w(0) = 0$, we must have $J = \emptyset$. This means $w(t) = 0$ for all $t$.

Recall the dynamics defined by 10 for the case $d = 1$ is

$$\frac{dw}{dt} = -\frac{\partial L_{\text{CP}}^D}{\partial w} = \sum_{i=1}^n \frac{x_i - y_i}{1 + e^{w(x_i - y_i)}}. \tag{17a}$$

$$\frac{dy_i}{dt} = -\frac{\partial L_{\text{CP}}^G}{\partial y_i} = \frac{w}{1 + e^{w(y_i - x_i)}}. \tag{17b}$$

Along the path $\theta(t) \in E$, we have $w(t)(y_i(t) - x_i(t)) = 0, \forall t$, thus

$$\frac{dy_i(t)}{dt} = \frac{0}{1 + e^0} = 0, \quad \forall i,$$

which implies $y_i(t) = y_i(0), \forall i$. This further implies $\frac{dw(t)}{dt} = \sum_i [x_i(t) - y_i(t)] = \sum_i [x_i(0) - y_i(0)] \triangleq \delta_0$. Since $\theta(0) \in E$ but not in $M$, thus we must have $\sum_i x_i(0) \neq \sum_i y_i(0)$, implying $\delta_0 \neq 0$. Thus $\frac{dw(t)}{dt} = \delta_0 \neq 0$, which contradicts $w(t) = 0, \forall t$. This contradiction implies that the original assumption that there is a set $M \subsetneq \bar{M} \subseteq E$ such that $\bar{M}$ is an invariant set does not hold. Thus the largest invariant subset of $E$ must be a subset of $M$.

Finally, if $\theta(0) \in M$, then the gradients are zero, thus $\theta(t) \in M, \forall t$, which implies that $M$ itself is a invariant set. Thus $M$ is indeed the largest invariant set of $E$.

### E.4 PROOF OF CLAIM 5.3

We restate the claim below.

**Claim E.1** *When $d = 1$, for the training dynamics of (10), at any point other than the desired solution $(y_1, \ldots, y_n) = (x_1, \ldots, x_n)$ in the set of stationary solutions, the Jacobian has an eigenvalue with positive real part* [3].

When $d = 1$, the dynamical system is given by

$$\frac{dw}{dt} = -\frac{\partial L_{\mathrm{CP}}^D}{\partial w} = \sum_{i=1}^{n} \frac{x_i - y_i}{1 + e^{w(x_i - y_i)}}. \tag{18a}$$

$$\frac{dy_i}{dt} = -\frac{\partial L_{\mathrm{CP}}^G}{\partial y_i} = \frac{w}{1 + e^{w(y_i - x_i)}}. \tag{18b}$$

We can write it as

$$\dot{\theta} = h(\theta) = (h_0(\theta), \ldots, h_n(\theta)),$$

where $h_i$ is the right hand side of the $i$-th equation. The Jacobian $J = \nabla h$ satisfies the following:

$$J_{11} = \frac{\partial h_0}{\partial w} = \sum_{i=1}^{n} \frac{(x_i - y_i)^2}{1 + e^{w(y_i - x_i)}},$$

$$J_{1,i+1} = \frac{\partial h_0}{\partial y_i} = -\frac{1}{1 + e^{w(x_i - y_i)}} + w(x_i - y_i)\frac{e^{w(y_i - x_i)}}{(1 + e^{w(y_i - x_i)})^2}, \forall i = 1, \ldots, n,$$

$$J_{i+1,1} = \frac{\partial h_i}{\partial w} = \frac{1}{1 + e^{w(y_i - x_i)}} + w(y_i - x_i)\frac{e^{w(y_i - x_i)}}{(1 + e^{w(y_i - x_i)})^2},$$

$$J_{i+1,i+1} == \frac{\partial h_i}{\partial y_i} = -\frac{w^2 e^{w(y_i - x_i)}}{(1 + e^{w(y_i - x_i)})^2}$$

$$J_{ij} = 0, \text{ if } i, j \geq 2 \text{ and } i \neq j.$$

When evaluated at $\theta = (0, y_1, \ldots, y_n)$, we have

$$J_{11} = \sum_{i=1}^{n} \frac{(x_i - y_i)^2}{2},$$

$$J_{1,i+1} = -\frac{1}{2}, \forall i = 1, \ldots, n,$$

$$J_{i+1,1} = \frac{1}{2},$$

$$J_{i+1,j+1} = 0, \quad i, j \geq 1.$$

---

[3]The stability of typical equilibria of smooth ODEs is determined by the sign of real part of eigenvalues of the Jacobian matrix. It is unstable if at least one eigenvalue has positive real part. See, e.g., Mescheder (2018) for the discussions.

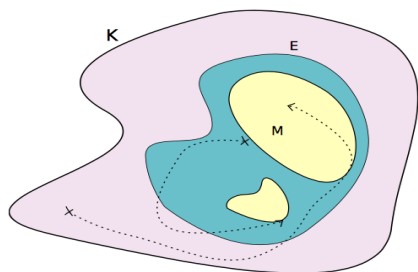

Figure 8: Graphical illustration of LaSalle's invariance principle. Figure from Cohen & Rouhling (2017). Starting from $K$, the dynamics will finally converge to the set $M$.

It is easy to verify that $J$ only has two non-zero eigenvalues

$$\lambda_{1,2} = \frac{1}{2}J_{11} \pm \frac{1}{\sqrt{2}}\sqrt{-1}.$$

Therefore, when $x_i = y_i, \forall x_i$, the eigenvalues of the Jacobian have zero real part. When some $x_i \neq y_i$, then the two eigenvalues of the Jacobian have a positive real part, meaning that this stationary point is not stable. This proves the result: in the set of stationary solutions $M = \{(w,y) : w = 0, \sum_i x_i = \sum_i y_i\}$, any point other than the desired solution $(y_1, \ldots, y_n)$ is not stable.

### E.5 ANALYSIS OF CP-GAN DYNAMICS FOR CONVEX CASE

Next, we consider the training dynamics of a generalized CP-GAN for a convex discriminator.

In (9), we used a log-linear discriminator $D(u) = \log(1 + e^{-w^T u})$ (although we called it a "linear discriminator" previously for simplicity). In this subsection, we show that the same analysis can be extended to a "convex-linear" discriminator. The empirical version with the convex-linear discriminator is presented below; it is an extension of (9).

$$\min_{\theta=w\in\mathbb{R}^d} L_{\mathrm{CP}}^D(Y;\theta) \triangleq \sum_{i=1}^n \phi(w^T(y_i - x_i)),$$

$$\min_{Y=(y_1,\ldots,y_n)\in\mathbb{R}^{d\times n}} L_{\mathrm{CP}}^G(Y;\theta) \triangleq \sum_{i=1}^n \phi(w^T(x_i - y_i)).$$

Consider the dynamics corresponding to the simultaneous gradient descent for CP-GAN:

$$\frac{dw}{dt} = -\frac{\partial L_{\mathrm{CP}}^D}{\partial w} = \sum_{i=1}^n (x_i - y_i)\phi'(w^T(y_i - x_i)), \tag{19a}$$

$$\frac{dy_i}{dt} = -\frac{\partial L_{\mathrm{CP}}^G}{\partial y_i} = w\phi'(w^T(x_i - y_i)) \tag{19b}$$

We still consider the function

$$V = \frac{1}{2}\|w\|^2 + \frac{1}{2}\sum_{i=1}^n \|x_i - y_i\|^2.$$

According to the following result, $V$ is non-increasing along the trajectory of CP-GAN training with convex discriminators, thus it is still a global Lyapunov function for the dynamics (19).

**Claim E.2** *Suppose $\phi$ is a convex function. Along the trajectory of the dynamics defined in Eq.* (19), *we have $\frac{dV(t)}{dt} \leq 0$.*

Proof of Claim E.2: Let $u_i = w^T(y_i - x_i), i = 1, \ldots, n$, we have

$$\frac{dV}{dt} = w^T \frac{dw}{dt} + \sum_i (y_i - x_i)^T \frac{dy_i}{dt} \tag{20a}$$

$$= w^T \sum_{i=1}^n (x_i - y_i)\phi'(w^T(y_i - x_i)) + \sum_i (y_i - x_i)^T w \phi'(w^T(x_i - y_i)) \tag{20b}$$

$$= \sum_{i=1}^n w^T(x_i - y_i)\phi'(w^T(y_i - x_i)) + \sum_{i=1}^n w^T(y_i - x_i)\phi'(w^T(x_i - y_i)) \tag{20c}$$

$$= \sum_{i=1}^n (-u_i)\phi'(u_i) + \sum_{i=1}^n u_i \phi'(-u_i) \tag{20d}$$

$$= \sum_{i=1}^n u_i[\phi'(-u_i) - \phi'(u_i)] \le 0, \tag{20e}$$

In the last step we used the fact that $u(\phi'(-u) - \phi'(u)) \le 0, \ \forall u \in \mathbb{R}$ due to the convexity of $\phi$. Note that in the above computation, we skip the time index $t$.

Thus $V$ is a Lyapunov function for the CP-GAN dynamics with convex discriminators. $\square$

It is not hard to extend the convergence results to this case, and we skip the details.

## F   PROOF OF PROPOSITION 2 FOR $n = 2$

Without loss of generality, we can consider two points $x_1 = 0, x_2 = 1$. The problem becomes

$$g^{\mathrm{CP}}(Y) = \sup_{f \in \mathcal{F}} \frac{1}{2} \log \frac{1}{1 + \exp(f(0) - f(y_1))} + \frac{1}{2} \log \frac{1}{1 + \exp(f(1) - f(y_2))},$$

where $\mathcal{F}$ is the set of continuous functions with domain $\mathbb{R}^d$. We compute all values of $g^{\mathrm{CP}}(Y)$ as follows.

**Case 1**: The two generated points are the same as the true data points, i.e., $\{x_1, x_2\} = \{y_1, y_2\}$. If $y_1 = 0, y_2 = 1$, then

$$g^{\mathrm{CP}}(Y) = \frac{1}{2}[\log 1/2 + \log 1/2] = -\log 2 \approx -0.6937.$$

If $y_1 = 1, y_2 = 0$, then

$$g^{\mathrm{CP}}(Y) = \sup_{f \in \mathcal{F}} \left[ \frac{1}{2} \log \frac{1}{1 + \exp(f(0) - f(1))} + \frac{1}{2} \log \frac{1}{1 + \exp(f(1) - f(0))} \right]$$

$$= \sup_{t \in \mathbb{R}} \left[ \frac{1}{2} \log \frac{1}{1 + \exp(t)} + \frac{1}{2} \log \frac{1}{1 + \exp(-t)} \right]$$

$$= -\log 2.$$

**Case 2**: Exactly one of the generated points coincides with the corresponding true data point, i.e., $|\{i : y_i = x_i\}| = 1$. Without loss of generality, we can check the case $y_1 = 0, y_2 \ne 1$. Then

$$g^{\mathrm{CP}}(Y) \ge \sup_{f \in \mathcal{F}} \frac{1}{2} \log \frac{1}{1 + \exp(f(0) - f(0))} + \frac{1}{2} \log \frac{1}{1 + \exp(f(1) - f(y_2))}$$

$$= -\frac{1}{2} \log 2 + \sup_{t \in \mathbb{R}} \frac{1}{2} \log \frac{1}{1 + \exp(t)}$$

$$= -\frac{1}{2} \log 2 \approx -0.3466.$$

The value is achieved when $f(1) - f(y_2) \to -\infty$ (or more precisely, there is a sequence of functions such that the difference $f(1) - f(y_2)$ goes to minus infinity).

**Case 3**: Both generated points are different from the true data points, i.e., $|\{i : y_i = x_i\}| = 1$. Then

$$
\begin{aligned}
g^{\mathrm{CP}}(Y) &\geq \sup_{f \in \mathcal{F}} \frac{1}{2} \log \frac{1}{1 + \exp(f(0) - f(y_1))} + \frac{1}{2} \log \frac{1}{1 + \exp(f(1) - f(y_2))} \\
&= \sup_{t_1 \in \mathbb{R}, t_2 \in \mathbb{R}} \frac{1}{2} \log \frac{1}{1 + \exp(t_1)} + \frac{1}{2} \log \frac{1}{1 + \exp(t_2)} \\
&= 0.
\end{aligned}
$$

The value is achieved when $f(1) - f(y_2) \to -\infty$ and $f(0) - f(y_2) \to -\infty$.

According to the above results, the only global minima are $\{y_1, y_2\} = \{x_1, x_2\}$. In addition, from any $Y$, it is easy to verify that there is a non-decreasing path from $Y$ to a global minimum.

# G   PROOF OF PROPOSITION 2 FOR GENERAL $n$

Recall the function

$$
g(Y) = \sup_f \frac{1}{n} \sum_{i=1}^{n} \log \frac{1}{1 + \exp(f(y_i) - f(x_i))} = - \inf_f \frac{1}{n} \sum_{i=1}^{n} \log\left(1 + \exp(f(y_i) - f(x_i))\right).
$$

**Proposition 5** *(restatement of Proposition 2 ) Suppose $x_1, x_2, \ldots, x_n \in \mathbb{R}^d$ are distinct. The global minimal value of $g^{\mathrm{CP}}(Y)$ is $-\log 2$, which is achieved iff $\{x_1, \ldots, x_n\} = \{y_1, \ldots, y_n\}$. Furthermore, the function $g(Y)$ has no sub-optimal basin.*

The key of the proof is to compute the values of $g(Y)$ for any $Y$. Roughly speaking, the value $g(Y)$ can be computed by the following process:

(1) We can build a graph with vertices representing distinct values in $x_i's, y_i's$ and draw directed edges from $x_i$ to $y_i$. This graph can be decomposed into cycles and trees.

(2) Each vertex in a cycle contributes $-\frac{1}{n} \log 2$ to the value $g(Y)$.

(3) Each vertex in a tree contributes $0$ to the value $g(Y)$.

Putting these ideas together, the value $g(Y)$ equals $-\frac{1}{n} \log 2$ times the number of vertices in the cycles.

The outline of this section is as follows. In the first subsection, as a warm-up example, we prove that if $\{y_1, y_2, \ldots, y_n\} = \{x_1, \ldots, x_n\}$, then $Y$ is a global minimum of $g(Y)$. In the second subsection, as another warm-up example, we prove that if there exists some $y_i \notin \{x_1, \ldots, x_n\}$, then it is not a global minimum. In the third subsection, we prove Proposition 2, in three steps. The proofs of some technical lemmas will be provided in the remaining subsections.

## G.1   WARM-UP EXAMPLE 1: ALL GENERATED POINTS MATCH THE TRUE POINTS

Prove that if $\{y_1, y_2, \ldots, y_n\} = \{x_1, \ldots, x_n\}$, then $Y$ is a global minimum of $g(Y)$.

Suppose $y_i = x_{\sigma(i)}$, then $(\sigma(1), \sigma(2), \ldots, \sigma(n))$ is a permutation of $(1, 2, \ldots, n)$. We view $\sigma$ as a mapping from $\{1, 2, \ldots, n\}$ to $\{1, 2, \ldots, n\}$. Pick an arbitrary $i$, then in the infinite sequence $i, \sigma(i), \sigma(\sigma(i)), \sigma^{(3)}(i), \ldots$ there exists at least two numbers that are the same. Suppose $\sigma^{(k_0)}(i) = \sigma^{(k_0+T)}(i)$ for some $k_0, T$, then since $\sigma$ is a one-to-one mapping we have $i = \sigma^{(T)}(i)$. Then we obtain a cycle $C = (i, \sigma(i), \sigma^{(2)}(i), \ldots, \sigma^{(T-1)}(i))$.

We can divide $\{1, 2, \ldots, n\}$ into the collection of finitely many cycles $C_1, C_2, \ldots, C_K$. Each cycle $C_k = (c_k(1), c_k(2), \ldots, c_k(m_k))$ satisfies $c_k(j+1) = \sigma(c_k(j)), j = 1, 2, \ldots, m_k$ where $c_k(m_k +$

1) is defined as $c_k(1)$. Now we calculate the value of $g(Y)$.

$$g(Y) = \sup_f \frac{1}{n} \sum_{i=1}^{n} \log \frac{1}{1 + \exp(f(y_i) - f(x_i)))}$$

$$\overset{(i)}{=} -\inf_f \frac{1}{n} \sum_{k=1}^{K} \sum_{i \in C_k} \log\left(1 + \exp(f(y_i) - f(x_i))\right)$$

$$= -\inf_f \frac{1}{n} \sum_{k=1}^{K} \sum_{j=1}^{m_k} \log\left(1 + \exp(f(x_{c_k(j+1)}) - f(x_{c_k(j)}))\right)$$

$$\overset{(ii)}{=} -\frac{1}{n} \sum_{k=1}^{K} \inf_f \sum_{j=1}^{m_k} \log\left(1 + \exp(f(x_{c_k(j+1)}) - f(x_{c_k(j)}))\right)$$

$$= -\frac{1}{n} \sum_{k=1}^{K} \inf_{t_1, t_2, \dots, t_{m_k} \in \mathbb{R}} \left[ \sum_{j=1}^{m_k-1} \log\left(1 + \exp(t_{j+1} - t_j)\right) + \log\left(1 + \exp(t_1 - t_{m_k})\right) \right]$$

$$\overset{(iii)}{=} -\frac{1}{n} \sum_{k=1}^{K} m_k \log(1 + \exp(0))$$

$$= -\log 2.$$

Here (i) is because $\{1, 2, \dots, n\}$ is the combination of $C_1, \dots, C_K$ and $i \in C_k$ means that $i = c_k(j)$ for some $j$. (ii) is because $C_k$'s are disjoint and $f$ can be any continuous function; more specifically, the values of $f$ at $x_i$'s for $i$ in two different cycles are independent, i.e., the choice of $\{f(x_i) : i \in C_{k_1}\}$ is independent of the choice of $\{f(x_i) : i \in C_{k_2}\}$ if $k_1 \neq k_2$, thus we can take the infimum over each cycle (i.e. put "inf" inside the sum over $k$). (iii) is because $\sum_{j=1}^{m-1} \log(1 + \exp(t_{j+1} - t_j)) + \log(1 + \exp(t_1 - t_m))$ is a convex function of $t_1, t_2, \dots, t_m$ and the minimum is achieved at $t_1 = t_2 = \dots t_m = 0$.

## G.2 WARM-UP EXAMPLE 2: NEW POINT GENERATED

Suppose $y_j \in \{x_1, \dots, x_n\}, \forall j$, and there exist some $x_{i_0}$ that is not equal to any $y_j$. In this part, we compute the value $g(Y)$ and show that $Y$ is not a global minimum. The computation for this example will illustrate how a "free" variable reduces the objective value $g(Y)$ by at least $-\frac{1}{n} \log 2$. Later in the general proof, we will see that any vertex not in a cycle will reduce the objective value by exactly $-\frac{1}{n} \log 2$.

Consider the term $\log(1 + \exp(f(y_{i_0}) - f(x_{i_0})))$. Since $x_{i_0}$ does not appear in any other term in $\sum_i \log(1 + \exp(f(y_i) - f(x_i)))$, the choice of $f(x_{i_0})$ is free. Therefore, no matter what values of $f(x_1), \dots, f(x_{i_0-1}), f(x_{i_0+1}), \dots, f(x_n)$ and $f(y_1), \dots, f(y_n)$ are, we can always pick $f(x_{i_0})$ so that $f(y_{i_0}) - f(x_{i_0}) \to -\infty$, making the term $\log(1 + \exp(f(y_{i_0}) - f(x_{i_0}))) \to 0$.

$$g(Y) = -\inf_f \frac{1}{n} \sum_{i=1}^{n} \log\left(1 + \exp(f(y_i) - f(x_i))\right)$$

$$= -\inf_f \frac{1}{n} \sum_{i \neq i_0} \log\left(1 + \exp(f(y_i) - f(x_i))\right) + 0$$

$$\geq -\frac{1}{n} \sum_{i \neq i_0} \log(1 + 1)$$

$$= -\frac{n-1}{n} \log 2.$$

## G.3 FORMAL PROOF OF PROPOSITION 2

This proof is divided into three steps. In Step 1, we compute the value of $g(Y)$ if all $y_i \in \{x_1, \dots, x_n\}$. This is the major step of the whole proof. In Step 2, we compute the value of $g(Y)$

for any $Y$. In Step 3, we show that if $Y$ is not a global minimum, then there is a non-decreasing continuous path from $Y$ to a global minimum.

### G.3.1 STEP 1: COMPUTE $g(Y)$ THAT ALL $y_i \in \{x_1, \ldots, x_n\}$

Assume

$$y_i \in \{x_1, \ldots, x_n\}, \forall i. \tag{21}$$

We build a directed graph $G = (V, A)$ as follows. The set of vertices $V = \{1, 2, \ldots, n\}$ represents $x_1, x_2, \ldots, x_n$. We draw a directed edge $(i, j) \in A$ if $y_i = x_j$; in this case, there is a term $\log(1 + \exp(f(x_j) - f(x_i)))$ in the expression of $g(Y)$. Note that in this directed graph, it is possible to have a self-loop $(i, i)$, which corresponds to the case $y_i = x_i$. Because of the assumption (21), we can simply express

$$
\begin{aligned}
g(Y) &= -\inf_f \frac{1}{n} \sum_{i=1}^{n} \log\left(1 + \exp(f(y_i) - f(x_i))\right) \\
&= -\inf_f \frac{1}{n} \sum_{(i,j) \in A} \log\left(1 + \exp(f(x_j) - f(x_i))\right).
\end{aligned} \tag{22}
$$

Each $y_i$ correspond to a unique $x_j$, thus the outdegree of $i$, denoted as outdegree$(i)$, must be exactly 1; in other words, for each $i$ there is exactly one directed edge going out from $i$. The indegree of each $i$, denoted as indegree$(i)$, can be any number in $\{0, 1, \ldots, n\}$.

To proceed, we need a few definitions from standard graph theory.

**Definition G.1** *(walk, path and cycle) In a directed graph $G$, a directed walk (or more simply, walk) $W = (v_0, e_1, v_1, e_2, \ldots, v_{m-1}, e_m, v_m)$ is a sequence of vertices and edges such that $v_i \in V$ for every $i \in \{0, 1, \ldots, m\}$ and $e_i$ is a directed edge from $v_{i-1}$ to $v_i$ for every $i \in \{1, \ldots, m\}$. The set of vertices in $W$ is denoted as $V(W)$, and the set of edged in $W$ is denoted as $A(W)$. If $v_0, v_1, \ldots, v_m$ are distinct we call it a directed path (or path), and we say the length of the path is $m$. If $v_0, v_1, \ldots, v_{m-1}$ are distinct and $v_m = v_0$, we call it a directed cycle (or cycle).*

Note that we always say $v$ has a path to itself $v$ (with length 0), no matter whether there is an edge between $v$ to itself or not. This is because the degenerate walk $W = (v)$ satisfies the conditions of the above definition.

**Definition G.2** *(tree) A directed tree is a directed graph $T = (V, A)$ with a designated node $r \in V$, the root, such that there is exactly one path from $v$ to $r$ for each node $v \in V$ and there is no edge from the root $r$ to itself. The depth of a node in a tree is the length of the path from the node to the root (define the depth of the root to be 0). A subtree of a directed graph $G$ is a subgraph $T$ of $G = (V, A)$ which is a directed tree. The set of vertices in $T$ is denoted as $V(T)$, and the set of edged in $T$ is denoted as $A(T)$.*

We prove a lemma that states that the graph can be decomposed into the union of cycles and trees. A graphical illustration is given in Figure 10.

**Lemma 2** *Suppose $G = (V, A)$ is a directed graph and outdegree$(v) = 1, \forall v \in V$. Then we have the following:*

*(a) There exist cycles $C_1, C_2, \ldots, C_K$ and subtrees $T_1, T_2, \ldots, T_M$ such that each edge $v \in A$ appears either in exactly one of the cycles or in exactly one of the subtrees.*

*(b) The root of each subtree $u_m$ is a vertex of a certain cycle $C_k$ where $1 \leq k \leq K$. In addition, each vertex of the graph appears in exactly one of the following sets: $V(C_1), \ldots, V(C_K), V(T_1) \backslash \{u_1\}, \ldots, V(T_M) \backslash \{u_M\}$,*

*(c) There is at least one cycle in the graph;*

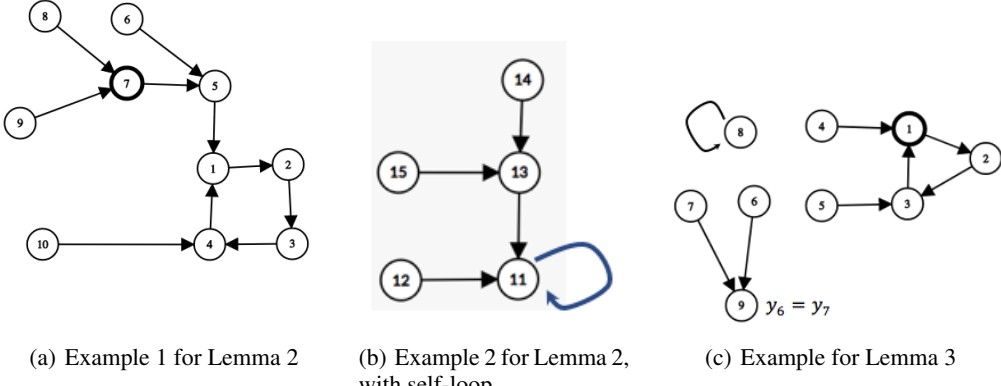

(a) Example 1 for Lemma 2     (b) Example 2 for Lemma 2,     (c) Example for Lemma 3
with self-loop

Figure 9: The first figure is a connected component of a graph. It contains $10$ vertices and $10$ directed edges. It can be decomposed into a cycle and two subtrees. The cycle consists of vertices $1, 2, 3, 4$. The first subtree consists of edge $(10, 4)$ and vertices $10, 4$, and the second subtree consists of edges $(8, 7), (9, 7), (7, 5), (6, 5), (5, 1)$. The second figure is another connected component of the same graph as the first figure. It has one cycle being a self-loop, and two trees attached to it. The third figure is an example of Lemma 3. There is one vertice $9$ with outdegree $0$, which corresponds to $y_6$ and $y_7$ since both $6$ and $7$ are connected to it. The fact that the two edges have the same head $9$ implies that $y_6 = y_7$.

With this lemma, we are ready to compute the value $g(Y)$. According to Lemma 2, we have

$$
\begin{aligned}
-ng(Y) &= \inf_f \sum_{i=1}^{n} \log\left(1 + \exp(f(y_i) - f(x_i))\right) \\
&= \inf_f \left[ \sum_{k=1}^{K} \sum_{i \in V(C_k)} \log\left(1 + \exp(f(y_i) - f(x_i))\right) + \sum_{m=1}^{M} \sum_{i \in V(T_m) \setminus \{u_m\}} \log\left(1 + \exp(f(y_i) - f(x_i))\right) \right] \\
&\geq \inf_f \left[ \sum_{k=1}^{K} \sum_{i \in V(C_k)} \log\left(1 + \exp(f(y_i) - f(x_i))\right) \right] \triangleq g_{\text{cyc}}.
\end{aligned}
$$

$$(23)$$

We then compute $g_{\text{cyc}}$. Since $C_k$ is a cycle, we have $X_k \triangleq \{x_i : i \in C_k\} = \{y_i : i \in C_k\}$. Since the cycles $C_k$'s are disjoint, we have $X_k \cap X_l = \emptyset, \forall k \neq l$, meaning that the values $f(x_i), f(y_i)$ for $i$ in one cycle $C_k$ are independent of the values corresponding to a different cycle $C_l$. Then the sum in the expression of $g_{\text{cyc}}$ can be decomposed according to different cycles.

$$
\begin{aligned}
g_{\text{cyc}} &= \inf_f \left[ \sum_{k=1}^{K} \sum_{i \in V(C_k)} \log\left(1 + \exp(f(y_i) - f(x_i))\right) \right] \\
&= \sum_{k=1}^{K} \inf_f \sum_{i \in V(C_k)} \log\left(1 + \exp(f(y_i) - f(x_i))\right)
\end{aligned}
$$

Similar to Step 1, we can show that the infimum for each cycle is achieved when the values $f(x_i) = f(x_j), \forall i, j \in V(C_k)$. A more detailed proof is given as follows. Pick an arbitrary $k$, and suppose all the edges of $C_k$ are $(v_1, v_2), (v_2, v_3), \dots, (v_{r-1}, v_r), (v_r, v_1)$, where $r = |V(C_k)|$ is the number

of vertices in the cycle $C_k$. Denote $v_{r+1} = v_1$. Then

$$
\begin{aligned}
&\inf_f \sum_{i \in V(C_k)} \log\left(1 + \exp(f(y_i) - f(x_i))\right) \\
&= \inf_f \sum_{j=1}^r \log\left(1 + \exp(f(x_{v_{j+1}}) - f(x_{v_j}))\right) \\
&= \inf_{t_1, t_2, \ldots, t_r \in \mathbb{R}} \left[\sum_{j=1}^{r-1} \log\left(1 + \exp(t_{j+1} - t_j)\right)\right) + \log\left(1 + \exp(t_1 - t_r)\right)\right] \\
&= r \log 2 = |V(C_k)| \log 2.
\end{aligned}
\tag{24}
$$

The infimum is achieved when $f(x_{v_1}) = \cdots = f(x_{v_r})$, or equivalently, $f(x_i) = f(x_j), \forall i, j \in V(C_k)$. Therefore,

$$
g_{\text{cyc}} = \log 2 \sum_{k=1}^K |V(C_k)|.
\tag{25}
$$

According to (23) and (25), we have

$$
-ng(Y) \geq \sum_{k=1}^K |V(C_k)| \log 2.
\tag{26}
$$

Next, we prove that for any $\epsilon > 0$, there exists a continuous function $f$ such that

$$
-ng(Y) < \sum_{k=1}^K |V(C_k)| \log 2 + \epsilon.
\tag{27}
$$

Let $N$ be a large positive number such that

$$
n \log\left(1 + \exp(-N)\right)) < \epsilon.
\tag{28}
$$

Pick a continuous function $f$ as follows.

$$
f(x_i) = \begin{cases} 0, & i \in \bigcup_{k=1}^K V(C_k), \\ N \cdot \text{depth}(i), & i \in \bigcup_{m=1}^M V(T_m). \end{cases}
\tag{29}
$$

Note that the root $u_m$ of a tree $T_m$ is also in a certain cycle $C_k$, thus the value $f(x_{u_m})$ is defined twice in (29), but in both definitions its value is 0, thus the definition is valid. For any $i \in V(C_k)$, suppose $y_i = x_j$, then both $i, j \in V(C_k)$ which implies $f(y_i) - f(x_i) = f(x_j) - f(x_i) = 0$. For any $i \in V(T_m)\backslash\{u_m\}$, suppose $y_i = x_j$, then by the definition of the graph $(i, j)$ is a directed edge of the tree $T_m$, which means that $\text{depth}(i) = \text{depth}(j) + 1$. Thus $f(y_i) - f(x_i) = f(x_j) - f(x_i) = -N$. In summary, for the choice of $f$ in (29), we have

$$
f(y_i) - f(x_i) = \begin{cases} 0, & i \in \bigcup_{k=1}^K V(C_k), \\ -N, & i \in \bigcup_{m=1}^M V(T_m). \end{cases}
\tag{30}
$$

For the choice of $f$ in (29), we have

$$
\begin{aligned}
&- nF(Y; f) \\
&= \sum_{i=1}^{n} \log\left(1 + \exp(f(y_i) - f(x_i))\right) \\
&= \left[\sum_{k=1}^{K} \sum_{i \in V(C_k)} \log\left(1 + \exp(f(y_i) - f(x_i))\right) + \sum_{m=1}^{M} \sum_{i \in V(T_m)\setminus\{u_m\}} \log\left(1 + \exp(f(y_i) - f(x_i))\right)\right] \\
&\overset{(30)}{=} \left[\sum_{k=1}^{K} \sum_{i \in V(C_k)} \log\left(1 + \exp(0)\right) + \sum_{m=1}^{M} \sum_{i \in V(T_m)\setminus\{u_m\}} \log\left(1 + \exp(-N)\right)\right] \\
&= \sum_{k=1}^{K} |V(C_k)| \log 2 + \sum_{k=1}^{M} (|V(T_m)| - 1) \log\left(1 + \exp(-N)\right) \\
&\leq \sum_{k=1}^{K} |V(C_k)| \log 2 + n \log\left(1 + \exp(-N)\right) \\
&\overset{(28)}{<} \sum_{k=1}^{K} |V(C_k)| \log 2 + \epsilon.
\end{aligned}
$$

This proves (27).

Combining (26) and (27), we have $-ng(Y) = \sum_{k=1}^{K} |V(C_k)| \log 2$, or equivalently,

$$
g(Y) = \frac{1}{n} \sum_{k=1}^{K} |V(C_k)| \log 2.
$$

### G.3.2 STEP 2: COMPUTE $g(Y)$ FOR ANY $Y$

In the general case, not all $y_i$'s lie in the set $\{x_1, \ldots, x_n\}$. Denote the set

$$
H = \{i : y_i \in \{x_1, \ldots, x_n\}\}, \quad H^c = \{j : y_j \notin \{x_1, \ldots, x_n\}\}.
$$

Since $y_j$'s in $H^c$ may be the same, we define the set of such distinct values of $y_j$'s as

$$
Y_{\text{out}} = \{y \in \mathbb{R}^d : y = y_j, \text{ for some } j \in H^c\}.
$$

Let $n_{\text{out}} = |Y_{\text{out}}|$, then there are total $n + n_{\text{out}}$ distinct values in $x_1, \ldots, x_n, y_1, \ldots, y_n$. Without loss of generality, assume $y_1, \ldots, y_{n_{\text{out}}}$ are distinct (this is because the value of $g(Y)$ does not change if we re-index $x_i$'s and $y_i$'s as long as the subscripts of $x_i, y_i$ change together), then

$$
Y_{\text{out}} = \{y_1, \ldots, y_{n_{\text{out}}}\}.
$$

We build a directed graph $G = (V, A)$ as follows. The set of vertices $V = \{1, 2, \ldots, n, n+1, \ldots, n+n_{\text{out}}\}$ represents $x_1, x_2, \ldots, x_n$ and $y_1, \ldots, y_{n_{\text{out}}}$. For $i, j \leq n$, we draw a directed edge $(i, j) \in A$ if $y_i = x_j$; in this case, there is a term $\log(1 + \exp(f(x_j) - f(x_i)))$ in the expression of $g(Y)$. For $1 \leq i \leq n$ and $1 \leq j \leq n_{\text{out}}$, we draw a directed edge $(i, n+j) \in A$ if $y_i = y_j$; in this case, there is a term $\log(1 + \exp(f(y_j) - f(x_i)))$ in the expression of $g(Y)$. Note that in this directed graph, it is possible to have a self-loop $(i, i)$, which corresponds to the case $y_i = x_i$.

This graph has the following property: for each $1 \leq i \leq n$, the outdegree is exactly 1; for each $n \leq j \leq n + n_{\text{out}}$, the outdegree of $j$ is 0. Moreover, each vertex representing some $y_i$ has an incoming degree at least 1 (since otherwise this vertex will not be created); note that an vertex representing $x_i$ may have incoming degree 0. We present a lemma which is an extension of the Lemma 2. The proof is given in Appendix G.5.

**Lemma 3** *Suppose $G = (V, A)$ is a directed graph and outdegree$(v) \leq 1, \forall v \in V$. Then we have the following:*

*(a) There exist cycles $C_1, C_2, \ldots, C_K$ and subtrees $T_1, T_2, \ldots, T_M$ such that each edge $v \in A$ appears either in exactly one of the cycles or in exactly one of the subtrees.*

*(b) Denote the root of subtree $T_m$ as $u_m$, $m = 1, \ldots, M$. Then $u_m$ is either a vertex of a certain cycle $C_k$ where $1 \le k \le K$, or a vertex with outdegree $0$.*

*(c) Suppose $u_1, \ldots, u_{M_0}$ are in a certain cycle, and $u_{M_0+1}, \ldots, u_M$ have outdegree $0$. Then each vertex of the graph appears in exactly one of the following sets: $V(C_1), \ldots, V(C_K), V(T_1)\backslash\{u_1\}, \ldots, V(T_{M_0})\backslash\{u_{M_0}\}, V(T_{M_0+1}), \ldots, V(T_M)$.*

Note that (b) is different from Lemma 2, because under the assumption of that lemma that each vertex has an outgoing edge, the root of a tree cannot have outdegree $0$.

The computation of $g(Y)$ is quite similar to the previous case. We will highlight a few small differences. We still have

$$
\begin{aligned}
&- ng(Y) \\
=& \inf_f \sum_{i=1}^n \xi(f(y_i) - f(x_i)) \\
=& \inf_f \left[ \sum_{k=1}^K \sum_{i \in V(C_k)} \xi(f(y_i) - f(x_i)) + \sum_{m=1}^M \sum_{i \in V(T_m)\backslash\{u_m\}} \xi(f(y_i) - f(x_i)) \right] \\
\ge& \inf_f \left[ \sum_{k=1}^K \sum_{i \in V(C_k)} \log\left(1 + \exp(f(y_i) - f(x_i))\right) \right] \\
\triangleq& g_{\text{cyc}}.
\end{aligned}
\tag{31}
$$

Same as before, we have $g_{\text{cyc}} = \sum_{k=1}^K |V(C_k)| \log 2$. Once we fix the values of $f(x_i)$ to be a constant for $i$ in cycles (which makes the first sum achieves the value $\sum_{k=1}^K |V(C_k)| \log 2$), we can always pick the values of $f$ on the vertices $j$ in the trees so that $f(y_j) - f(x_j) \to \infty$. Therefore, we have

$$
-ng(Y) = g_{\text{cyc}} = \sum_{k=1}^K |V(C_k)| \log 2.
$$

### G.3.3 STEP 3: FINDING A NON-DECREASING PATH TO A GLOBAL MINIMUM

Finally, we prove that for any $Y$, there is a non-decreasing continuous path from $Y$ to one global minimum $Y^*$. In other words, there is a continuous function $\eta : [0, 1] \to \mathbb{R}^{d \times n}$ such that $\eta(0) = \bar{Y}, \eta(1) = Y^*$ and $g(\eta(t))$ is a non-decreasing function with respect to $t \in [0, 1]$. In this proof, we will just describe the path in words, and skip the rigorous definition of the continuous function $\eta$, since it should be clear from the context how to define $\eta$.

The following claim shows that we can increase the value of $Y$ incrementally; see the proof in Appendix G.6

**Claim G.1** *For an arbitrary $Y$ that is not a global minimum, there exists another $\hat{Y}$ and a non-decreasing continuous path from $Y$ to $\hat{Y}$ such that $g(\hat{Y}) - g(Y) \ge \frac{1}{n} \log 2$.*

For any $Y$ that is not a global minimum, we apply Claim G.1 for finitely many times (no more than $n$ times), then we will arrive at one global minimum $Y^*$. We connect all non-decreasing continuous paths, and get a non-decreasing continuous path from $Y$ to $Y^*$. This finishes the proof of Proposition 2.

### G.4 PROOF OF LEMMA 2

We first prove a few observations. We will slightly extend the definition of "walk" to allow a walk with infinite length.

**Observation 1**: Suppose in a directed graph $G = (V, A)$, any vertex has outdegree exactly 1. Starting from any vertex $v_0 \in V(G)$, there is a unique walk with infinite length

$$W(v_0) \triangleq (v_0, e_1, v_1, e_2, v_2, \ldots, v_i, e_i, v_{i+1}, e_{i+1}, \ldots),$$

where $e_i$ is an edge in $A(G)$ with tail $v_{i-1}$ and head $v_i$.

Proof of Observation 1: At each vertex $v_i$, there is a unique outgoing edge $e_i = (v_i, v_{i+1})$ which uniquely defines the next vertex $v_{i+1}$. Continue the process, we have proved Observation 1. $\square$

**Observation 2**: Suppose in a directed graph $G = (V, A)$, any vertex has outdegree exactly 1. Suppose the unique walk starting from $v_0$ is $W(v_0) \triangleq (v_0, e_1, v_1, e_2, v_2, \ldots, v_i, e_i, v_{i+1}, e_{i+1}, \ldots)$, then the walk can be decomposed into two parts

$$W_1(v_0) = (v_0, e_1, v_1, e_2, v_2, \ldots, v_{i_0-1}, e_{i_1}, v_{i_0}),$$
$$W_2(v_0) = (v_{i_0}, e_{i_0+1}, v_{i_0+1}, e_{i_0+2}, v_{i_0+2}, \ldots),$$

where $W_1(v_0)$ is a directed path from $v_0$ to $v_{i_0}$ (i.e. the vertices $v_0, v_1, \ldots, v_{i_0}$ are distinct), and $W_2(v_0)$ is the repetition of a certain cycle (i.e. there exists $T$ such that $v_{i+T} = v_i$, for any $i \geq i_0$). This decomposition is unique.

**Proof of Observation 2**: Since there are only finitely many vertices in the graph, then some vertices must appear at least twice in $W(v_0)$. Among all such vertices, suppose $u$ is the one that appears the earliest in the walk $W(v_0)$, and the first two appearances are $v_{i_0} = u$ and $v_{i_1} = u$ and $i_0 < i_1$. Denote $T = i_1 - i_0$. Since there is a unique edge going out from any vertex, thus $v_{i_0+1}$ must be the same as $v_{i_1+1} = v_{i_0+1+T}$. Continue the process, we have $v_i = v_{i+T}$ for any $i \geq i_0$. Thus starting from $u = v_{i_0}$, the walk $W_2(v_0)$ will be repetitions of the cycle consisting of vertices $v_{i_0}, v_{i_0+1}, \ldots, v_{i_1-1}$, and we denote this cycle as $C_{k_0}$.

If the vertices before $v_{i_1}$ are not distinct, then there are at least two vertices $v_j = v_l$ where $0 \leq j < l \leq i_0$. This contradicts the definition of $i_0$. Therefore, $W_1(v_0)$ is a directed path from $v_0$ to $v_{i_0}$. $\square$

In Observation 2, for any $v_0 \in V$, the "transition point" $v_{i_0}$ in the infinite walk $W(v_0)$ is unique. We define the "first-touch-vertex" of $v_0$ to be $v_{i_0}$. The first-touch-vertex $u = v_{i_0}$ has the following properties: (i) $u \in C_k$ for some $k$; (ii) there exists a path from $v$ to $u$; (iii) any paths from $v$ to any vertex in the cycle $C_k$ other than $u$ must pass $u$. In other words, if an ant starts from $v_0$ and crawls along the edges, the first vertex in a cycle that it touches is $u$. Since any $v_0$ corresponds to a unique $W(v_0)$, its first-touch-vertex must exist and unique. Note that if $u$ is in some cycle, then its first-touch-point is $u$ itself.

As a corollary of Observation 2, there is at least one cycle in the graph. Suppose all cycles of $G$ are $C_1, C_2, \ldots, C_K$. Because the outdegree of each vertex is 1, these cycles must be disjoint, i.e., $V(C_i) \cap V(C_j) = \emptyset$ and $A(C_i) \cap A(C_j) = \emptyset$, for any $i \neq j$.

Define $R$ to be the set of vertices of $C_1, \ldots, C_m$ with indegree at least two, i.e.,

$$R \triangleq \{v : v \in C_k \text{ for some } k, \text{ and indegree}(v) \geq 2\}.$$

Suppose the elements of $R$ are $u_1, \ldots, u_M$. We denote the set of vertices in the cycles as

$$V_c = \bigcup_{k=1}^{K} V(C_1) \cup \cdots \cup V(C_K). \tag{32}$$

We describe the intuition behind the partitioning of $G$. Based on Observation 2, starting from any vertex outside of $V_c$ there is a unique path that reaches $V_c$. Combining all vertices that reach the cycles at $u_m$ (denoted as $V_m$), and the paths from these vertices to $u_m$, we obtain a directed subgraph $T_m$, which is connected with $V_c$ only via the vertex $u_m$. The subgraphs $T_m$'s are disjoint from each other since they are connected with $V_c$ via different vertices. In addition, each vertex outside of $V_c$ lies in exactly one of the subgraph $T_m$. Thus, we can partition the whole graph into the union of the cycles $C_1, \ldots, C_K$ and the subgraphs $T_1, \ldots, T_M$.

Next, we provide more formal definitions and proofs. For any $m \in \{1, 2, \ldots, M\}$, define

$$\bar{V}_m = \{v \in V(G) : u_m \text{ is the first-touch-vertex of } v\}, \tag{33a}$$

$$V_m = \bar{V}_m \backslash \{u_m\}, \tag{33b}$$

$$A_m = \{e \in A(G) : \text{ the tail of } e \text{ is in } V_m\}, \tag{33c}$$

$$T_m = (\bar{V}_m, A_m) \text{ is a subgraph of } G. \tag{33d}$$

We first show that $V_1, V_2, \ldots, V_M$ and $V(C_1), \ldots, V(C_K)$ form a partition of the edge set of $G$, i.e.,

$$V(G) = \left( \bigcup_{k=1}^{K} V(C_k) \right) \bigcup \left( \bigcup_{m=1}^{M} V_m \right), \tag{34a}$$

$$V_m \cap V_c = \emptyset, \quad \forall m, s,. \tag{34b}$$

$$V_m \cap V(C_k) = \emptyset, \quad \forall m, k. \tag{34c}$$

$$V(C_k) \cap V(C_l) = \emptyset, \quad \forall k \neq l, \tag{34d}$$

For any vertex $v \notin \left( \bigcup_{k=1}^{K} V(C_k) \right)$, its first-touch-vertex $w$ must be different from $v$ (otherwise $v$ would be in a certain cycle). Since $w$ is in a cycle, denoted as $C_{k_0}$, there is a directed edge with head $w$; in addition, since $w$ is a first-touch-vertex of $v \notin C_{k_0}$, then there must be another edge with head $w$. Thus the indegree of $w$ is at least 2, which means $w \in R$. Thus $w = u_m$ for some $m$, which implies $v \in V_m$. Since any $v \notin \left( \bigcup_{k=1}^{K} V(C_k) \right)$ must belong to $V_m$ for some $m$, we have proved (34a).

Since each vertex $v$ has a unique first-touch-vertex, thus a vertex cannot lie in two different sets $V_m$ and $V_s$, which proves (34b). Assume (34c) does not hold, i.e., there exists some $v \in V(C_k) \cap V_m$ for some $k, m$. Since the first-touch-vertex of $v$ is $v$ itself, according to (33a) we have $v = u_m$; but according to (33b) $v = u_m \notin V_m$, a contradiction. Thus (34c) holds. (34d) is obvious.

Next, we show that the edge sets of $T_1, \ldots, T_m$ and $C_1, \ldots, C_K$ form a partition of the edge set of $G$, i.e.,

$$A(G) = \left( \bigcup_{k=1}^{K} A(C_k) \right) \bigcup \left( \bigcup_{m=1}^{M} A(T_m) \right), \tag{35a}$$

$$A(T_m) \cap A(T_s) = \emptyset, \quad \forall m, s,. \tag{35b}$$

$$A(T_m) \cap A(C_k) = \emptyset, \quad \forall m, k. \tag{35c}$$

$$A(C_k) \cap A(C_l) = \emptyset, \quad \forall k \neq l, \tag{35d}$$

These relation can be proved easily by (34) and the definitions (33c) and (33d). For any edge $e_1 \in A(G) \backslash \left( \bigcup_{k=1}^{K} A(C_k) \right)$, we want to show that $e_1$ lies in at least one $T_m$. Suppose $e_1 = (v_0, v_1)$ where $v_0, v_1 \in V(G)$. Since $e_1$ is not in a cycle, $v_0 \notin V_c$, thus by (34a) we have $v_0 \in V_m$ for a certain $m$, which further implies $e_1 \in A_m = A(T_m)$ by the the definitions (33c) and (33d). Thus we proved (35a). Any edge $e$ in $A(T_m)$ has a tail $u \in V(T_m)$, thus according to (34b) we know $u \notin V_s, \forall s \neq m$. By the definition (33c), we have $e \notin A_m = A(T_m)$, which proves (35b). Similarly, any edge $e$ in $A(C_k)$ has a tail $u \in V(C_k)$, thus according to (34c) we know $u \notin V_m, \forall m$, thus we have $e \notin A_m = A(T_m)$, which proves (35c). The last one (35d) holds because the cycles are disjoint.

Finally, we prove that for any $m \in \{1, 2, \ldots, M\}$, $T_m$ is a subtree of $G$ with the root $u_m$. For any vertex $v_0$ in the subgraph $T_m$, consider the walk $W(v_0)$. Any path starting from $v_0$ must be part of $W(v_0)$. Starting from $v_0$ there is only one path from $v_0$ to $u_m$ which is $W_1(v_0)$, according to Observation 2. Therefore, by the definition of directed tree, $T_m$ is a directed tree with the root $u_m$.

## G.5 PROOF OF LEMMA 3

We utilize the result of Lemma 2 to prove Lemma 3. More specifically, we will reduce to the case of Lemma 2.

We build a new graph as follows. Denote the set of vertices in $G$ that have outdegree 0 as $D_0$. For any vertex in $D_0$, we add an edge from $D_0$ to itself (a self-loop), and obtain a new graph $\tilde{G} = (V, \tilde{A})$. Now each vertice in $\tilde{G}$ has outdegree exactly 1.

Each self-loop is a length-1 cycle, thus by adding edges to vertices in $D_0$ we have created $|D_0|$ length-1 cycles, denoted as $C_{K+1}, \ldots, C_{K+|D_0|}$. Suppose all other cycles are $C_1, C_2, \ldots, C_K$.

According to Lemma 1, there exist cycles $C_1, C_2, \ldots, C_K, C_{K+1}, \ldots, C_{K+|D_0|}$ and subtrees $T_1, T_2, \ldots, T_M$ such that: (a) Each edge $v \in \tilde{A}$ appears either in exactly one of the cycles or in exactly one of the subtrees. (b) The root of each subtree $u_m$ is a vertex of a certain cycle $C_k$ where $1 \leq k \leq K + |D_0|$. In addition, each vertex of the graph appears in exactly one of the following sets: $V(C_1), \ldots, V(C_K), V(T_1) \backslash \{u_1\}, \ldots, V(T_M) \backslash \{u_M\}$,

Now we remove the edges added to vertices in $D_0$ and restore the original graph. Then we have removed the cycles $C_{K+1}, \ldots, C_{K+|D_0|}$ and do not change other cycles and subtrees. We do change the relations between the trees and the cycles: previously, the root of each tree is a vertex in a certain cycle; now, since some cycles are removed, the roots of some trees will be a vertex with outdegree 0. Suppose these trees are $T_{M_0+1}, T_{M_0+2}, \ldots, T_M$.

After removing the edges added to vertices in $D_0$, $C_1, C_2, \ldots, C_K$ and subtrees $T_1, T_2, \ldots, T_M$ satisfy the following: (a) Each edge $v \in A$ appears either in exactly one of the cycles or in exactly one of the subtrees. (b) Denote the root of subtree $T_m$ as $u_m$, $m = 1, \ldots, M$. Then $u_m$ is either a vertex of a certain cycle $C_k$ where $1 \leq k \leq K$, or a vertex with outdegree 0. (c) Each vertex of the graph appears in exactly one of the following sets: $V(C_1), \ldots, V(C_K), V(T_1) \backslash \{u_1\}, \ldots, V(T_{M_0}) \backslash \{u_{M_0}\}$, $V(T_{M_0+1}), \ldots, V(T_M)$.

## G.6    PROOF OF CLAIM G.1

We first prove the case for $d \geq 2$. Suppose the corresponding graph for $Y$ is $G$, and $G$ is decomposed into the union of cycles $C_1, \ldots, C_K$ and trees $T_1, \ldots, T_m$. We perform the following operation: pick an arbitrary tree $T_m$ with the root $u_m$. We claim that there must be an edge $e$ with the head $u_m$. If $u_m > n$ (i.e. $u_m$ represents some $y_i$), then there must be an incoming edge to $u_m$, thus the claim holds. If $u_m \leq n$ (i.e. $u_m$ represents some $x_i$), then $u_m$ must have an outgoing edge, and this edge must be in a cycle (otherwise $u_m$ cannot be the root). Thus $u_m$ is chosen to be in a tree $T_m$ must because $u_m$ is a head of an edge.

Suppose $v$ is the tail of the edge $e$. Now we remove the edge $e = (v, u_m)$ and create a new edge $e' = (v, v)$. The new edge corresponds to $y_v = x_v$. The old edge $(v, u_m)$ corresponds to $y_v = x_{u_m}$ (and a term $\xi(f(x_{u_m}) - f(x_v))$) if $u_m \leq n$ or $y_v = y_{u_m-n} \notin \{x_1, \ldots, x_n\}$ (and a term $\xi(f(y_{u_m-n}) - f(x_v))$) if $u_m > n$. This change corresponds to the change of $y_v$: we change $y_v = x_{u_m}$ (if $u_m \leq n$) or $y_v = y_{u_m-n}$ (if $u_m > n$) to $\hat{y}_v = x_v$. Let $\hat{y}_i = y_i$ for any $i \neq v$.

Previously $v$ is in a tree $T_m$, now $v$ is the head of a new tree, and also part of the new cycle $C_{K+1} = (v, e', v)$. In this new graph, the number of vertices in cycles increases by 1, thus the value of $g$ increases by $-\frac{1}{n} \log 2$, i.e., $g(\hat{Y}) - g(Y) = \frac{1}{n} \log 2$.

Since $d \geq 2$, we can find a path in $\mathbb{R}^d$ from a point to another point without passing any of the points in $\{x_1, \ldots, x_n\}$. In the continuous process of moving $y_v$ to $\hat{y}_v$, the function values will not change except at the end that $y_v = x_v$. Thus there is a non-increasing path from $Y$ to $\hat{Y}$, in the sense that along this path the function values of $g$ does not decrease.

The illustration of this proof is given as below.

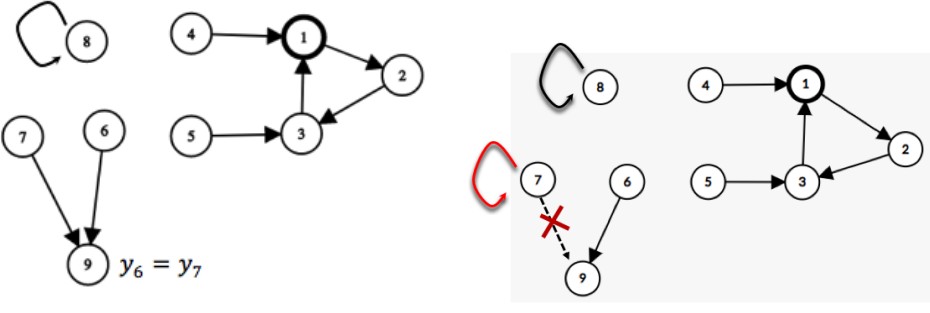

(a) Original graph          (b) Modified graph, with improved function value

Figure 10: Illustration of the proof of Claim G.1. For the figure on the left, we pick an arbitrary tree with the head being vertex 9, which corresponds to $y_6 = y_7$. We change $y_7$ to $\hat{y}_7 = x_7$ to obtain the figure on the right. Since one more cycle is created, the function value increases by $-\frac{1}{n}\log 2$.

For the case $d = 1$, the above proof does not work. The reason is that the path from $y_v$ to $\hat{y}_v$ may touch other points in $\{x_1, \ldots, x_n\}$ and thus may change the value of $g$. We only need to make a small modification: we move $y_v$ in $\mathbb{R}$ until it touches a certain $x_i$ that corresponds to a vertex in the tree $T_m$, at which point a cycle is created, and the function value increases by at least $\frac{1}{n}\log 2$. This path is a non-decreasing path, thus the claim is also proved.

## H    GENERATED SAMPLES IN SIMULATION

Here we show some samples of generated data in the appendix.

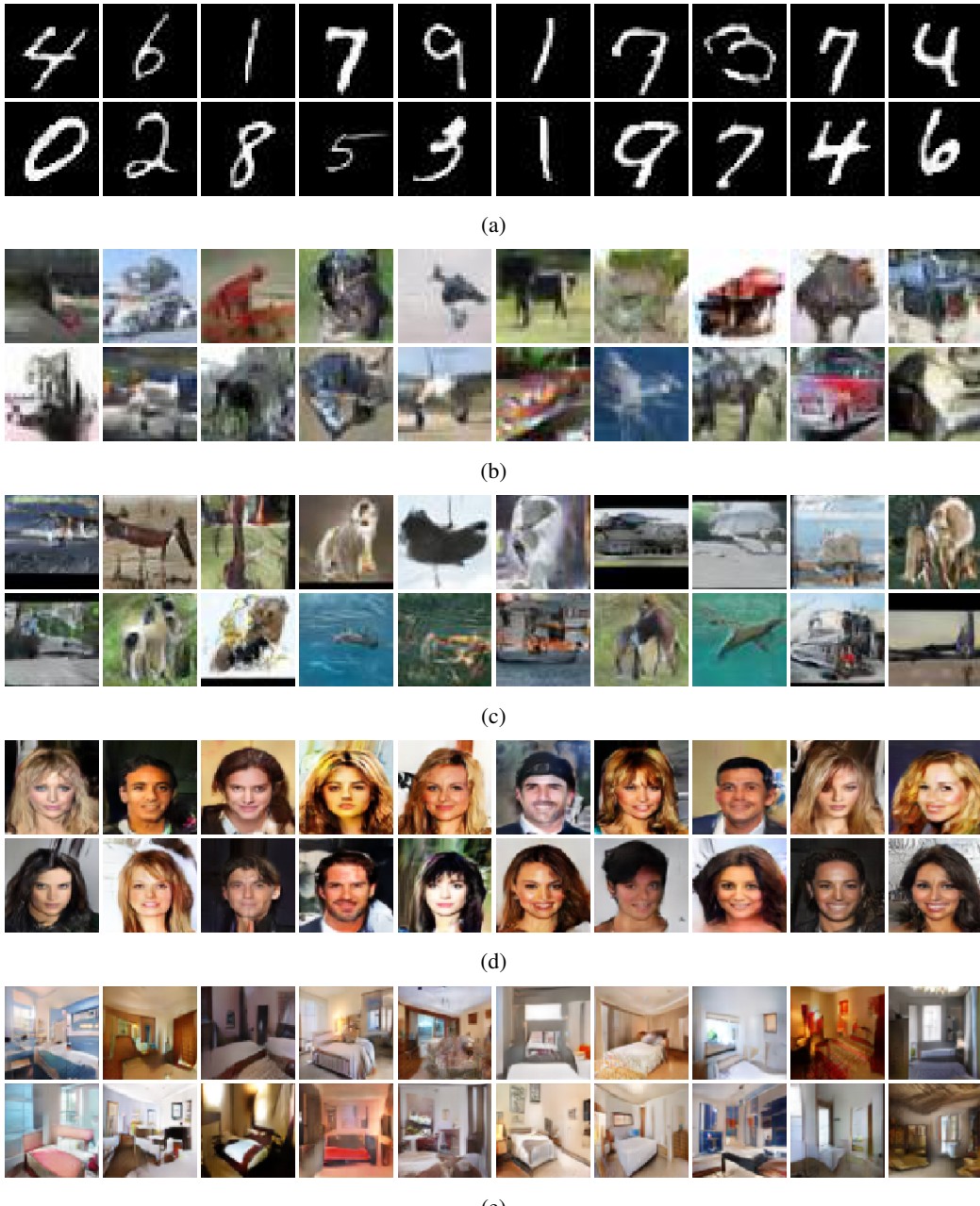

Figure 11: Generated (a)MNIST (b) CIFAR-10 (c) CelebA (d) LSUN samples by CPGAN.

