# OpenReview forum: "CP-GAN: Towards a Better Global Landscape of GANs"
_ICLR.cc/2020/Conference — Reject_

### Official Review · AnonReviewer3 · 2019-10-18
**Official Blind Review #3**

**Rating:** 8

**Review:**

In this paper, the authors introduce a new training loss for GANs.  This loss allows the outer optimization problem to have no spurious local minima, under an appropriate finite sample analysis.  In contrast, the authors establish that there are exponentially many spurious local minima under the conventional GAN training loss.  Under a linear discriminator model, the authors show that a standard GAN can not escape from collapsed modes in a finite sample analysis, whereas the new trining loss allows for such an escape (due to the presence of a Lyapunov functional with favorable properties).  The authors use this new training loss to train GANS on MNIST, CIFAR10, CelebA, and LSUN datasets, and observe mild improvements in Inception Scores and Frechet Inception Distances of the resulting generated images.

I recommend the paper be accepted because it provides a new formulation for training GANs that both demonstrates improved empirical performance while also allowing theoretically favorable properties (on spurious local minima and avoidance of mode collapse) that specifically do not hold for a standard GAN.

The primary question I am left with after reading the paper is: is there a probabilistic interpretation of the new loss function (equation 4a).  The authors justify this formulation because it allows analysis via Lyapunov functions, but it would be very useful to know if it itself is the maximum likelihood estimate under an alternate data model.  Such an explanation would improve the understandability of this method.

Minor comment:

The fourth bullet point under the contributions section should specific the sense in which the new GAN "performs better"


**Experience Assessment:**

I have read many papers in this area.

**Review Assessment: Checking Correctness Of Derivations And Theory:**

I assessed the sensibility of the derivations and theory.

**Review Assessment: Checking Correctness Of Experiments:**

I assessed the sensibility of the experiments.

**Review Assessment: Thoroughness In Paper Reading:**

I read the paper at least twice and used my best judgement in assessing the paper.

---

> ### Author Response · Authors · 2019-11-15
> **Our response to reviewer 3**
>
> Reviewer 3:
> We thank the reviewer for the detailed comments and support. Below, we provide detailed responses to each concern and question.
>
> Comment 1: The primary question I am left with after reading the paper is: is there a probabilistic interpretation of the new loss function (equation 4a).  The authors justify this formulation because it allows analysis via Lyapunov functions, but it would be very useful to know if it itself is the maximum likelihood estimate under an alternate data model.  Such an explanation would improve the understandability of this method.
>
> Response 1:
> Thanks for the comment.  See more discussions in Sec 2.2 in revised paper.
>      The JS-GAN is using binary classification loss as the “shell”, and a probabilistic interpretation is that p(x_i) is the probabilities of x_i being in class 1. Another interpretation is that D wants to find a single hyperplane that separates {x_i}, {y_i}, and -log(1 + exp(-x_i)) and -log(1 + exp(y_i)) mimics the hinge loss. However, the final goal is not binary classification, but generating y_i’s to fool D. We do not need to use a single hyperplane to separate them. Instead, we can have multiple hyperplanes.
>
>
> Comment 2: The fourth bullet point under the contributions section should specific the sense in which the new GAN "performs better"
>
> Response 2:
> Thanks for the comment. We add “in terms of Inception Scores and FID”.

---

### Official Review · AnonReviewer2 · 2019-10-22
**Official Blind Review #2**

**Rating:** 3

**Review:**

The paper attempts to perform global analysis of GAN on the issue of sub-optimal strict local minima and mode collapse, and proposes a new GAN formulation (CoupleGAN) that enjoys nice global properties. The paper is overall well written and conveys an interesting new formulation of GANs. However, the reviewer is concerned with the following questions:
The paper is mainly on analyzing the case when the true data has n points instead of on a continuous support. It would be more interesting to see theoretical guarantee on even Gaussian mixture model. Also since GANs are mostly known for generalizing what is seen to generate new data, whether converging only to the n points are good or not still worth debating.
In claim 4.2 and 4.3, what if the initialization of y is completely random? Then the claim cannot say anything on mode collapse. So is the formulation in the paper the real characterization of mode collapse?


**Experience Assessment:**

I have read many papers in this area.

**Review Assessment: Checking Correctness Of Derivations And Theory:**

I assessed the sensibility of the derivations and theory.

**Review Assessment: Checking Correctness Of Experiments:**

I assessed the sensibility of the experiments.

**Review Assessment: Thoroughness In Paper Reading:**

I read the paper at least twice and used my best judgement in assessing the paper.

---

> ### Author Response · Authors · 2019-11-15
> **Our response for Reviewer2**
>
> We thank the reviewer for the detailed comments. The comments lead us to add Appendix A to explain the motivation in detail, and make the paper stronger. We really appreciate the comments.
> Below, we provide detailed responses to each concern and question.
>
> ----------------
> Comment 1: “The paper is mainly on analyzing the case when the true data has n points instead of on a continuous support. It would be more interesting to see theoretical guarantee on even Gaussian mixture model.“
>
> Response 1: Thank you very much for the insightful comment.
> Short reply: (1) We add Appendix A to explain the motivation in much detail (1.5 pages with 6 figures), to elaborate our previous point “n-point mimics n-modes”. In particular, we highlight the "macro-learning" perspective.
>   (2) Multi-Gaussian is probably a more difficult problem than ours, and may rely on some techniques of this paper. It is an interesting future work.
>
>
> Longer reply:
> 1)The n-point model mimics the n-Gaussian model. It captures the macro-learning behavior. Consider learning a two-mode distribution P, starting from an initial two-mode distribution Q. There are two differences between P and Q: first, the locations of the two modes are different; second, the distribution within each mode is different. To learn the distribution, we want to eliminate both differences: first, move the two modes of Q to roughly overlap with the two modes of P which we call "``macro-learning"; second, adjust the distributions of each mode to match those of P, which we call ``"micro learning". Our analysis captures the macro-learning part.
>
> 2)The n-point model generalizes the 1-point model in Mescheder et al. ICML’18. Our analysis is already much more general than the 1-point model.
>
> 3) As R1 pointed out, there is a paper on learning a single Gaussian, but we did not notice an extension to 2-Gaussians yet. We think our analysis can be combined with 1710.10793 for future analysis of multi-Gaussian. Currently, this paper on the n-point case is already 34 pages.
>      We kindly remind the reviewer the current optimization theory for GAN is quite rare. Even the 2-point case was not proved for global convergence before (to our knowledge).
>
> --------------------------------
> Comment 2: Also since GANs are mostly known for generalizing what is seen to generate new data, whether converging only to the n points are good or not still worth debating.
>
> Response 2:
> Thank you for this comment. We add Appendix A.1 to explain. To summarize A.1:
> (1) Generalization is proved in a classical work Arora et al'17, and can be easily extended to our setting. More specifically, for JS-GAN and other GANs, the generalization error of "fitting n data points" is bounded in Arora et al'17. If the reviewer insists, we can even add a proof of generalization bound for CP-GAN in the final version (this is probably just simple exercise adopted from Arora et al.'17).
> (2) We provide some intuition about why it generalizes, by using a figure.
> (3) Anyways, fitting the training data is what GAN is doing in practice, and is also what neural-nets doing for image classification. However, fitting can be difficult and requires analysis. More broadly, generalization and optimization are orthogonal issues.
>
> --------------------------------
> Comment 3: In claim 4.2 and 4.3, what if the initialization of y is completely random? Then the claim cannot say anything on mode collapse. So is the formulation in the paper the real characterization of mode collapse?
>
> Response 3:
> First, if the initialization of y is random, then it depends on the gaps between y_i and y_j. Due to randomness, there could be one close pair (y_i, y_j) at initialization and stay close throughout, which causes mode collapse. If the random y turns out to be spread out (e.g. choose y_1 in [-1,0] and choose y_2 in [10,11]), then it either gets close to “mode collapse” and then stuck, or it never gets close to mode collapse and converges to global-min.
>      "Random" is hard to control (in GAN with neural-nets, we do not control Y directly). In our experiments of JS-GAN, we see some "random" initial point causes mode collapse and some causes success. With a better global landscape, the training shall be more stable to initialization.
>
> Second, as a prediction of our theory, for mode collapse, discriminator gradients are small compared to the successful cases. We did experiments on 2-Gaussian and 5-Gaussian data, and see the phenomenon.
>       As reviewed in Appendix B, existing works conjectured "improper loss function" and "weak discriminator" cause mode collapse. There is no rigorous definition of the cause. Given the context, we think we provided a more concrete hypothesis for future study.
>
> Third, characterizing mode collapse as bad local-min is just a partial goal of our paper. Our major motivation is to get a global landscape for further theoretical analysis.

---

### Official Review · AnonReviewer1 · 2019-10-23
**Official Blind Review #1**

**Rating:** 3

**Review:**

Authors propose a  modification to the original GAN formulation, by coupling the generated samples and the true samples to avoid mode collapse.

I have some concerns about the analysis and the experiments of the paper: Most of the analysis is tailored for a very simple linear discriminator case which for the WGAN means just matching the first moments. Even in this simple setup, they consider d=1 (the scalar case). I am not sure how one can generalize this analysis to a more realistic case. Also the experimental gains seem incremental which makes me worried about such generalization. Finally, there are a few works in the literature about understanding the optimization landscape of GANs. For a sample, see https://arxiv.org/abs/1706.04156 and https://arxiv.org/abs/1710.10793. The later uses a Lyp function to analysis the global convergence of a GAN.  Also there is a few papers about the mode collapse issue in GANs. See for example https://arxiv.org/abs/1712.04086


**Experience Assessment:**

I have published in this field for several years.

**Review Assessment: Checking Correctness Of Derivations And Theory:**

I assessed the sensibility of the derivations and theory.

**Review Assessment: Checking Correctness Of Experiments:**

I assessed the sensibility of the experiments.

**Review Assessment: Thoroughness In Paper Reading:**

I made a quick assessment of this paper.

---

> ### Author Response · Authors · 2019-11-15
> **Our response to reviewer 1**
>
> We thank the reviewer for the detailed comments. These comments are very helpful for improving the paper. The comments lead us to add Appendix B to explain related work. We really appreciate the comments.
> Below, we provide detailed responses.
>
> -------------------------------------------------
> Q1: Most of the analysis is tailored for a very simple linear discriminator case which for the WGAN means just matching the first moments.
>
> A1: First, we would like to kindly remind the reviewer that most of the proofs (10 pages, page 23-34) are for the powerful-D case.
>
> Second, while an ideal result is to analyze practical cases with neural-net discriminators, such an analysis for GANs is rare (if any). Most current analyzes are for linear-D. We provide results on both extremes: powerful-D and linear-D. We believe this provides convincing evidence regarding the nice properties of CP-GAN.
>
> Third, our analysis can be easily extended to a convex case for CP-GAN. We added Appendix E.5 to provide details. Note that the analysis of WGAN-GP is not our focus, and we just showed a small negative result for it. At a high level, the analysis is one part of the big picture, trying to validate the benefit of CP-GAN.
>
> -------------------------------------------------
> Q2: Even in this simple setup, they consider d=1 (the scalar case). I am not sure how one can generalize this analysis to a more realistic case.
>
> A2: Please note that we are not only considering d=1:
>
> For general d, we proved the existence of a global Lyapunov function for CP-GAN and a negative result for JS-GAN and WGAN-GP. This differentiates CP-GAN from other GANs.
>
> For general d, we have proved global convergence to the set of critical points of the Lyapunov function (but did not state explicitly). To present this more explicitly, we changed Proposition 3 to add a convergence result for general d. Currently, there is a technical difficulty for proving convergence to the set of stationary points for general d, and it is left as future work.
>
> We think our results may be generalizable to GANs with neural-nets. The reason is that overparameterization analysis in recent advances such as https://arxiv.org/abs/1806.07572 relies heavily on a “shell problem” with a nice landscape. We provided such a “shell problem”.
>
> ----------------------------------------------------------------------------------------
> Q3: Also the experimental gains seem incremental which makes me worried about such generalization.
>
> A3: With only two lines of code change in PyTorch, we improve FID scores by 16 points on CIFAR10 over JS-GAN and 25 points on STL10 (also a few points better than WGAN-GP). We think this experimental gain is remarkable compared to the algorithmic changes. We think this is a very promising experimental result.

---

> ### Author Response · Authors · 2019-11-15
> **Our response to reviewer 1**
>
>
> Q4: Finally, there are a few works in the literature about understanding the optimization landscape of GANs. For a sample, see https://arxiv.org/abs/1706.04156 and https://arxiv.org/abs/1710.10793. The later uses a Lyp function to analysis the global convergence of a GAN.  Also there is a few papers about the mode collapse issue in GANs. See for example https://arxiv.org/abs/1712.04086
>
> A4: Thanks a lot for pointing out these references. We add detailed discussions in the Appendix B “Related Work”. They help a lot in positioning our work in the context. They are mostly complementary, and may lead to interesting new works when combined with our analysis. We summarize the contents of Appendix B here.
>
> [R1] https://arxiv.org/abs/1706.04156: This work is cited in our paper. It only analyzed local convergence. Moreover, as pointed out by Mescheder et al.’18, “the assumption of absolute continuity is not true for common use cases of GANs, where both distributions may lie on lower dimensional manifolds”. This is why [R1] proved local convergence, while Mescheder et al.’18 proved that even for single point the local convergence does not hold.  The fundamental difference is that [R1] considers the micro-learning effect of "letting density change continuously", while Mescheder et al.’18 considers the macro-learning effect of "letting a single mode move".
>
> [R2] https://arxiv.org/abs/1710.10793 Thank you for pointing out this paper. It looks very interesting, and we have explained in detail the relation to this paper in the revised version.
>     --It is a very different paper. The major difference with our paper is that they considered the single-mode case, while we consider the multi-mode case (a simplified version; see detailed discussions in Appendix A). There are a few other differences: (a) They consider the population version, and we consider the empirical version. (b) They consider the quadratic discriminator, and we analyze both the powerful discriminator case and the linear discriminator.
>     --It is complementary to ours. This paper and ours capture two somewhat orthogonal aspects of the problem. Our comment in the revised paper is: “To extend to the multi-mode case such as multi-Gaussian, as we discussed earlier, there is a macro-learning effect and micro-learning effect. Our work on n-point distributions captures the macro-learning effect, and Feizi et al. captures the micro-learning effect for Gaussian data. In the future, it would be quite interesting to combine the analysis of Feizi et al. and our analysis to the multi-Gaussian case.”   (To be more rigorous, we think [R2] captures both the micro-learning studied in [R1] and macro-learning studied in Mescheder et al.’18; but that is just the single-mode macro-learning, and we study multi-mode macro-learning. Single-mode learning is somehow easy according to Mescheder et al.’18, so the major challenge of [R2] may be to capture the micro-learning effect. Anyhow, we would read [R2] more carefully later, to make the claim more precise.).
>
>     It is quite interesting that [R2] also used a Lyapunov function. However, the underlying mathematics of [R2] and our paper are quite different. The formulation of this paper (16) is a matrix factorization version of a bi-linear zero-sum game (19). Our formulation involves logarithmic (and extendable to convex), and is a non-zero sum game.
>
> Mode collapse and [R3] https://arxiv.org/abs/1712.04086.  Most papers on mode collapse are empirical; [R3] has a rigorous theory. We discuss in detail the differences in revised paper.
>       First,  they did not provide theoretical analysis for a specific GAN; in contrast, we prove theoretical results of  specific JS-GAN and CP-GAN formulations. Second, we provided an explanation for ``why mode collapse happens'', by linking mode collapse
> to a fundamental optimization subject ``bad basin''.  Third, their focus is to mitigate ``bad basin'', and our starting point is to analyze the global landscape, and the link to mode collapse is a natural byproduct of  the analysis.
>        The major difference is: they analyze in the "statistical distance level, andn proves TV(P^m, Q^m) is better than TV(P,Q)", and borrow the insight to use packing. We directly analyze the GAN min-max problem (or game), not analyzing a general distribution distance like [R3].

---

### Author Response · Authors · 2019-11-15
**Meta-response: modification of papers**

Dear reviewers,

Thank you for your effort and time.

We mainly added the following parts in the revised paper to address the comments:
   1) Add Appendix A (with 4 figures), to explain why learning n-points can be viewed as the "macro-learning" part of learning n-modes.
   2) Add Appendix B on some related works. Especially elaborate a fundamental work on learning single-Gaussian (single-mode). Will include more in the final version.
   3) Other parts:
       --add Appendix A.1 to briefly discuss generalization;
       --add interpretation of the loss in Sec 2.2;
       --add E.5 to extend dynamical analysis to convex-linear D.

---

### Decision · Program_Chairs · 2019-12-19

**Decision:**

Reject

**Comment:**

The paper is proposed a rejection based on majority reviews.